# Beyond First-order Asymptotics in Sequential Mean Testing

**Vikas Deep** [1]  **Shubhada Agrawal** [2]

## Abstract

We revisit the problem of sequentially testing the mean of bounded distributions in a level-$\alpha$ power-one framework. We study a $\mathrm{KL}_{\inf}$-based sequential test that is known to attain the information-theoretic lower bound on the expected stopping time with exact constants as $\alpha \to 0$. Going beyond first-order asymptotics, we establish a central limit theorem (CLT) for the stopping time of this test. Our analysis proceeds in two steps. First, we prove a novel CLT for the $\mathrm{KL}_{\inf}$ statistic itself, characterizing its fluctuations around its deterministic limit. We then leverage this result to show that the stopping time, centered appropriately and scaled by $\sqrt{\log(1/\alpha)}$, converges in distribution to a Gaussian limit with an explicit variance. This yields a second-order characterization of an asymptotically optimal sequential test for bounded distributions. Finally, we present numerical experiments that corroborate our theoretical findings.

## 1. Introduction

Sequential mean testing is a fundamental primitive underlying online experimentation, adaptive monitoring, and sequential learning. Informally, with streaming data, we want to decide as quickly as possible whether the mean is at a target level $m_o$, while controlling false alarms at level $\alpha \in (0, 1)$.

This problem arises in applications such as A/B testing and adaptive monitoring, where samples are costly, decisions are made online, and the data distribution is rarely parametric. Motivated by this, we consider the setting of the sequential mean testing problem where the underlying data distribution is non-parametric, with support bounded in $[0, 1]$. Specifically, we observe an i.i.d. sequence $X_1, X_2, \ldots$ with a common probability distribution $p$ supported on $[0, 1]$ and mean $m(p) := \mathbb{E}_p[X]$. Given $m_o \in (0, 1)$, we study the classical sequential mean testing problem

$$H_0 : m(p) = m_o \qquad \text{versus} \qquad H_1 : m(p) \neq m_o$$

in an $\alpha$-correct power-one framework.

Given $\alpha > 0$, an $\alpha$-*correct (or level $\alpha$) power-one sequential test* (initially studied by Darling & Robbins (1967) for parametric models) controls type I error at level $\alpha$, while attaining power one under the alternative. Such tests are specified by a stopping time $\tau_\alpha$, where stopping corresponds to rejecting $H_0$. Formally, if $\mathcal{P}$ and $\mathcal{Q}$ denote the sets of distributions in $H_0$ and $H_1$, then $\tau_\alpha$ satisfies

$$\sup_{p \in \mathcal{P}} p(\tau_\alpha < \infty) \leq \alpha, \quad \text{and} \quad \inf_{q \in \mathcal{Q}} q(\tau_\alpha < \infty) = 1.$$

Recently, Agrawal & Ramdas (2025) developed a general theory of $\alpha$-correct power-one sequential tests for composite hypotheses, establishing tight information-theoretic lower bounds and matching upper bounds on the *expected stopping time* $\mathbb{E}_q[\tau_\alpha]$ in two different asymptotic regimes. In the small-error regime $\alpha \downarrow 0$, for the bounded nonparametric mean testing problem, their lower bounds and optimal tests are in terms of a certain KL-projection function, termed $\mathrm{KL}_{\inf}$ (defined in Section 3.2). Moreover, these tests satisfy the first-order optimality:

$$\lim_{\alpha \to 0} \frac{\mathbb{E}_q[\tau_\alpha]}{\log(1/\alpha)} = \frac{1}{\mathrm{KL}_{\inf}(q, m_o)},$$

for every fixed alternative $q$ with $m(q) \neq m_o$.

However, the expected stopping time alone does not tell the full story: sequential procedures can exhibit substantial run-to-run variability, and in practice one often cares about predictability guarantees, such as the probability that the test terminates by a given deadline. For example, in sequential monitoring of safety metrics (e.g., extreme losses or excess risk in financial systems), a procedure with a small $\mathbb{E}[\tau_\alpha]$ may still be unsatisfactory if it has a non-negligible probability of stopping very late. Such delayed detection can incur high costs in safety-critical applications. These considerations motivate a second-order, distributional analysis of the stopping time of sequential tests.

---

[1]Department of Information Systems and Analytics at the National University of Singapore [2]Department of Electrical Communication Engineering (ECE) at the Indian Institute of Science, Bengaluru. Correspondence to: Vikas Deep <vikas_deep@nus.edu.sg>, Shubhada Agrawal <shubhada@iisc.ac.in>.

*Proceedings of the 43$^{rd}$ International Conference on Machine Learning*, Seoul, South Korea. PMLR 306, 2026. Copyright 2026 by the author(s).

In this work, *we establish a Central Limit Theorem (CLT) for the stopping time* of an optimal sequential test for the nonparametric mean-testing problem. The CLT for $\tau_\alpha$ quantifies typical fluctuations around its deterministic growth rate, yielding explicit variance and a Gaussian approximation that refines first-order asymptotic optimality into a more complete characterization of stopping-time uncertainty.

CLTs for stopping times are well understood in the classical setting of first-passage times for random walks (Gut, 2009; Asmussen, 2003). We refer the reader to Section 2 for a detailed literature review. These tools underpin stopping-time CLTs for parametric likelihood-ratio procedures, which are known to yield asymptotically optimal $\alpha$-correct, power-one tests for sequential mean testing in parametric models (Robbins & Siegmund, 1974). This approach works for the parametric models because the log-likelihood ratio statistic in these settings can be expressed as a sum of i.i.d. log-likelihood increments, thereby inducing a fixed random-walk structure. However, this fails in the non-parametric settings.

The nonparametric $\mathrm{KL}_{\mathrm{inf}}$ statistic is defined through an optimization over distributions (equivalently, via a dual optimization problem), and the resulting process cannot be reduced to a fixed random walk in a straightforward manner. As a result, classical first-passage CLTs for random walks do not directly apply, and existing techniques do not appear to readily yield distributional limit theorems for stopping times for the optimal $\mathrm{KL}_{\mathrm{inf}}$-based power-one tests. Further, to the best of our knowledge, no stopping-time CLT is known for asymptotically optimal nonparametric power-one sequential tests.

The main contributions of this work are as follows.

1. We establish a CLT for an appropriate centering and scaling of the empirical $\mathrm{KL}_{\mathrm{inf}}$-statistic, characterizing its fluctuations around its deterministic limit under bounded nonparametric models (Theorem 4.2).

2. Building on the above, we derive the first stopping-time CLT for an asymptotically optimal nonparametric power-one sequential test based on $\mathrm{KL}_{\mathrm{inf}}$ statistic, obtaining an explicit Gaussian limit for the properly normalized stopping time (Theorem 4.4).

3. As an application, we construct an asymptotically valid confidence interval for the stopping time when $\alpha$ is small, using only a *single simulation run* (Proposition 4.5).

4. We validate the theory through simulations (Section 5) as well as experiments on real-world DSSAT crop-yield dataset (https://dssat.net).

Our key idea is to leverage the fact that the dual represen-

tation of $\mathrm{KL}_{\mathrm{inf}}$ is a one-dimensional convex optimization problem, which rewrites the statistic as the maximum of a concave objective indexed by a single scalar parameter. We show that the empirical maximizer computed from the data converges to the maximizer of the dual problem for the true (unknown) distribution at $O_p(1/\sqrt{n})$ rate. Crucially, once the maximizer is stable, the optimization itself does not contribute to the leading-order fluctuations, leading to the Gaussian limit in Theorem 4.2.

To pass from CLT for $\mathrm{KL}_{\mathrm{inf}}$-statistic to the stopping-time CLT, we use the fact that the stopping time grows on the order of $\log(1/\alpha)$. Around that scale, the accumulated statistic behaves like a deterministic linear drift plus the fluctuation term identified above. A standard boundary-crossing inversion argument then converts fluctuations of the accumulated statistic into fluctuations of the time at which the boundary is crossed, giving a Gaussian limit for the properly rescaled stopping time with an explicit variance constant inherited from the first step (Theorem 4.4).

A key technical step in our approach is justifying the CLT for the empirical-$\mathrm{KL}_{\mathrm{inf}}$ statistic at the stopping time $\tau_\alpha$. This corresponds to verifying an Anscombe-type condition (see Anscombe (1952) and Gut (2009, Chapter 1)), ensuring that the fixed-$n$ CLT transfers to a CLT at the stopping time $\tau_\alpha$. Standard proofs for this typically rely on i.i.d. increments and Kolmogorov's maximal inequality (Gut, 2009, Chapter 1). However, since the empirical-$\mathrm{KL}_{\mathrm{inf}}$ statistic does not directly yield an i.i.d. increments formulation, a Taylor expansion decomposes the statistic into a sum of a leading partial-sum term with i.i.d. increments, and a smaller remainder term (see Lemma A.8).

**Paper organization.** Section 2 reviews related work on sequential testing and stopping-time limit theorems. Section 3 introduces the bounded nonparametric mean testing model, defines $\mathrm{KL}_{\mathrm{inf}}$ and its dual representation, and describes the $\mathrm{KL}_{\mathrm{inf}}$-based stopping rule. Section 4 presents our main results, including a CLT for the empirical $\mathrm{KL}_{\mathrm{inf}}$ statistic and the resulting stopping time $\tau_\alpha$, with proofs deferred to the appendices. Section 4.2 contains the results related to application of stopping time CLT to construct confidence interval for stopping time. Section 5 reports numerical experiments that support the theoretical findings, and we conclude with a discussion of future directions.

## 2. Related Work

Our results connect two classical strands of work: (i) *sequential testing* for mean and, (ii) distributional limit theorems for *stopping times* (in particular, first-passage times). We briefly review the most relevant literature in each area below.

**Sequential testing.** Sequential testing originated in Wald's framework with the sequential probability ratio test (SPRT),

which is optimal for simple-vs-simple parametric hypotheses: among all tests satisfying given error constraints, it minimizes the expected number of samples (Wald, 1992; Wald & Wolfowitz, 1948). Further, power-one and $\alpha$-correct sequential tests were first studied by Darling & Robbins (1967).

In classical parametric settings, the optimality of sequential tests is characterized by a certain Kullback-Leibler (KL) divergence; see early optimality results and lower bounds in sequential design and testing (Siegmund, 2013; Chernoff, 1959). In nonparametric settings (eg, mean testing with bounded support), the relevant information quantity becomes a KL-projection onto the null set, often denoted as a $\mathrm{KL}_{\mathrm{inf}}$. This KL-projection function has previously appeared in the lower bounds in the multi-armed bandit literature, dating back to works of Lai & Robbins (1985); Burnetas & Katehakis (1996), and bandit algorithms designed using an empirical version of it have been shown to be optimal in a wide variety of non-parametric bandit settings (Honda & Takemura, 2010; 2015; Agrawal et al., 2020; 2021a;b; Jourdan et al., 2022). Furthermore, this KL-projection function has previously appeared in the construction of asymptotically optimal confidence intervals for the mean of a distribution and the average treatment effect in A/B testing (Deep et al., 2024; 2025).

Recently, Agrawal & Ramdas (2025) develop a general theory of power-one, level-$\alpha$ sequential tests for composite hypotheses. However, they focus on the expected stopping time. In contrast, the focus of this work is to understand other properties of these optimized tests.

**Stopping-time limit theorems and first-passage CLTs.** Distributional limit theorems for stopping times are classical in applied probability and sequential analysis, with a large literature devoted to first-passage times of drifted random walks and their refinements via renewal theory. These results yield both laws of large numbers and central limit theorems for stopping times (Gut, 2009; Asmussen, 2003). In parametric settings, log-likelihood ratio statistics (and their mixture or generalized variants) evolve additively, allowing first-passage CLTs to be applied directly to obtain distributional approximations for stopping times. Another independent work, Mukhopadhyay (2020) provides sufficient conditions for establishing a CLT for a stopping time defined through a test statistic, assuming that the test statistic itself already satisfies a CLT. Establishing such a CLT for our test statistic is itself one of the main technical contributions of this work. Furthermore, our stopping rule does not directly fit into their framework, preventing a black-box application of their results.

## 3. Preliminaries: Setup and Background

We now formally describe our setup. Let $\mathcal{B}$ denote the set of all probability measures supported on $[0, 1]$. For any $p \in \mathcal{B}$, write $m(p) := \mathbb{E}_p[X]$ for its mean. We observe an i.i.d. sequence $X_1, X_2, \ldots \in [0, 1]$ with common law $p \in \mathcal{B}$, and define the natural filtration $\mathcal{F}_n := \sigma(X_1, \ldots, X_n)$. We fix $m_o \in (0, 1)$ and an error tolerance level $\alpha \in (0, 1)$. Throughout, $\mathrm{KL}(q, p)$ denotes the Kullback-Leibler divergence (with the usual convention $\mathrm{KL}(q, p) = +\infty$ if $q \ll p$). We use the notation $\mathbb{P}_p(\cdot)$ to denote the probability of the input event under the measure on an infinite length sequence $X_1, X_2, X_3, \ldots$, generated i.i.d. from $p$. Similarly, we now use $\mathbb{E}_p[\cdot]$ to denote the expectation under this joint measure.

We consider the following mean testing problem in this work,

$$
\begin{aligned}
H_0 &: p \in \mathcal{P}^{\mathrm{bd}} := \{p \in \mathcal{B} : m(p) = m_o\}, \\
H_1 &: p \in \mathcal{Q}^{\mathrm{bd}} := \{p \in \mathcal{B} : m(p) \neq m_o\}.
\end{aligned}
\tag{1}
$$

### 3.1. $\alpha$-correct Power-one Sequential Tests

A sequential test is specified by a stopping time $\tau_\alpha$ (with respect to $(\mathcal{F}_n)_{n \geq 1}$) taking values in $\mathbb{N} \cup \{\infty\}$, with the convention that the procedure rejects $H_0$ at time $\tau_\alpha$ if $\tau_\alpha < \infty$; otherwise, the test never rejects.

**Definition 3.1** ($\alpha$-correct power-one sequential test)**.** For $\alpha \in (0, 1)$, a stopping time $\tau_\alpha$ is called an $\alpha$-*correct power-one sequential test* for (1) if

$$
\sup_{p \in \mathcal{P}^{\mathrm{bd}}} p(\tau_\alpha < \infty) \leq \alpha,
$$

$$
\inf_{q \in \mathcal{Q}^{\mathrm{bd}}} q(\tau_\alpha < \infty) = 1.
$$

To formally introduce the known asymptotically optimal sequential test, we first introduce the $\mathrm{KL}_{\mathrm{inf}}$ function, which will be used in the construction of the test.

### 3.2. KL Projection Function: $\mathrm{KL}_{\mathrm{inf}}$

$\mathrm{KL}_{\mathrm{inf}}$ for bounded distributions is a well-known object in the literature (Honda & Takemura, 2010; 2015). For a given $q \in \mathcal{B}$ and $m_o \in (0, 1)$,

$$
\mathrm{KL}_{\mathrm{inf}}(q, m_o) \triangleq
\begin{cases}
\mathrm{KL}_{\mathrm{inf}}^+(q, m_o), & m_o \geq m(q), \\
\mathrm{KL}_{\mathrm{inf}}^-(q, m_o), & m_o < m(q),
\end{cases}
$$

with

$$
\mathrm{KL}_{\mathrm{inf}}^+(q, m_o) := \inf_{p \in \mathcal{B}:\, m(p) \geq m_o} \mathrm{KL}(q, p),
$$

$$
\mathrm{KL}_{\mathrm{inf}}^-(q, m_o) := \inf_{p \in \mathcal{B}:\, m(p) \leq m_o} \mathrm{KL}(q, p).
$$

Note that $\mathrm{KL}_{\mathrm{inf}}(q, \cdot)$ is an increasing function on $[m(q), 1]$ and decreasing on $[0, m(q)]$.

**Dual representation.** While the (primal) definition of $\mathrm{KL}_{\mathrm{inf}}$ involves an optimization over the space of probability measures, which can be computationally inconvenient, $\mathrm{KL}_{\mathrm{inf}}$ is, by now, a well-understood object. In particular, it has a dual formulation that provides a tractable and explicit characterization that is especially convenient for algorithmic and analytical purposes. In our proofs, we will move between the primal and dual, exploiting properties of both the problems. We therefore recall below the dual representations:

$$\mathrm{KL}_{\mathrm{inf}}^+(q, m_o) = \sup_{\lambda \in \left[0, \frac{1}{1-m_o}\right]} \mathbb{E}_q\left[\log\left(1 - \lambda(X - m_o)\right)\right],$$
$$(2)$$
$$\mathrm{KL}_{\mathrm{inf}}^-(q, m_o) = \sup_{\lambda \in \left[-\frac{1}{m_o}, 0\right]} \mathbb{E}_q\left[\log\left(1 - \lambda(X - m_o)\right)\right].$$

**Dual maximizers.** Recall from Honda & Takemura (2010) that there exists a unique dual maximizer in (2). We denote by $\lambda^\star(q)$ the value of $\lambda$ that achieves the supremum in the definition of $\mathrm{KL}_{\mathrm{inf}}(q, m_0)$. If $m_o \geq m(q)$, then $\lambda^*(q)$ denotes the value of $\lambda$ that achieves the supremum in the definition of $\mathrm{KL}_{\mathrm{inf}}^+(q, m_o)$. Specifically, for $m_o \geq m(q)$,

$$\lambda^\star(q) \in \arg\max_{\lambda \in \left[0, \frac{1}{1-m_o}\right]} \mathbb{E}_q\left[\log\left(1 - \lambda(X - m_o)\right)\right].$$

Similarly, if $m_o < m(q)$, then $\lambda^*(q)$ denotes the corresponding maximizer in $\mathrm{KL}_{\mathrm{inf}}^-(q, m_o)$, and one can symmetrically define the maximizer for this case. Whenever it is clear from the context, we will drop the dependence of this maximizer on $q$, and instead refer to it as $\lambda^*$.

### 3.3. $\mathrm{KL}_{\mathrm{inf}}$- based Optimal Sequential Test

Now, we describe the optimal sequential test using the empirical-$\mathrm{KL}_{\mathrm{inf}}$ statistic, which is well known in the literature. Later, we will prove the CLT for the stopping time of this test.

For $x \in [0, 1]$ let $\delta_x$ denote a unit point mass at $x$. For $n \geq 1$, let

$$\hat{q}_n := \frac{1}{n} \sum_{i=1}^{n} \delta_{X_i}$$

denote the empirical distribution. Define

$$\tau_\alpha = \inf\left\{n \in \mathbb{N}: \, n \, \mathrm{KL}_{\mathrm{inf}}(\hat{q}_n, m_o) \geq \beta(n, \alpha)\right\}, \quad (3)$$

where $\beta(n, \alpha) = 1 + \log\left(\frac{2(1+n)}{\alpha}\right)$. Using the martingale construction proposed in Agrawal et al. (2021b, Lemma F.1), it is easy to see that $\tau_\alpha$ is an $\alpha$-correct stopping rule. Furthermore, for any $q \in \mathcal{Q}^{\mathrm{bd}}$,

$$\lim_{\alpha \to 0} \frac{\mathbb{E}_q[\tau_\alpha]}{\log(1/\alpha)} = \frac{1}{\mathrm{KL}_{\mathrm{inf}}(q, m_o)}.$$

The above equality (lower bound + upper bound) was implicitly proven in the multi-armed bandit literature (Agrawal et al., 2020; Jourdan et al., 2022), and later, made explicit in Agrawal & Ramdas (2025).

It is worth noting that the asymptotically optimal stopping rule $\tau_\alpha$ defined above tracks the empirical analogue $n \, \mathrm{KL}_{\mathrm{inf}}(\hat{q}_n, m_o)$, which behaves like an accumulated evidences against $H_0$, and compares it to a boundary $\beta(n, \alpha)$ to ensure $\alpha$-level type-I error control.

Our goal in this paper is to study finer distributional properties of the associated stopping time $\tau_\alpha$.

## 4. Main Results

In this section, we will establish a CLT for $\tau_\alpha$ as $\alpha \downarrow 0$. Towards this, we will first establish a CLT for the empirical $\mathrm{KL}_{\mathrm{inf}}(\hat{q}_n, m_o)$ statistic, which is the key technical contribution of the paper and is of independent interest.

For $I_{m_o} := [-1/m_o, 1/(1 - m_o)]$, $\lambda \in I_{m_o}$, and $x \in [0, 1]$, define

$$\ell(\lambda, x) := \log(1 - \lambda(x - m_o)).$$

Then, $(1 - \lambda(x - m_o)) \geq 0$. Further, it is 0 only if either $(\lambda, x) = (1/(1 - m_o), 1)$ or $(\lambda, x) = (-1/m_o, 0)$. This observation will be useful in the proofs of our results.

Honda & Takemura (2010) show that $\mathrm{KL}_{\mathrm{inf}}(\hat{q}_n, m_o) \to \mathrm{KL}_{\mathrm{inf}}(q, m_o)$ as $n \to \infty$ almost surely (as $\hat{q}_n \Rightarrow q$ almost surely, and $\mathrm{KL}_{\mathrm{inf}}(\cdot, m_o)$ is continuous in the weak topology). Thus, we will center the statistic around $\mathrm{KL}_{\mathrm{inf}}(q, m_o)$ in our CLT.

We now state a minor technical condition. It excludes only distributions with sufficiently heavy mass near the boundary point 1, while still allowing distributions that place positive probability mass at 1. In Appendix B, we show that this assumption is mild and holds for several commonly studied bounded-support distributions, including Bernoulli distributions, the uniform distribution, and most Beta distributions.

**Assumption 4.1.** For $q \in \mathcal{B}$ and $m_o \in (0, 1)$ with $m_o \neq m(q)$, assume the following: if $m_o > m(q)$ and $\mathbb{E}_q[(1 - m_o)/(1 - X)] = 1$, then $\mathbb{E}_q\left[1/(1 - X)^2\right] < \infty$; and if $m_o < m(q)$ and $\mathbb{E}_q[m_o/X] = 1$, then $\mathbb{E}_q\left[1/X^2\right] < \infty$.

**Theorem 4.2.** *Fix $q \in \mathcal{Q}^{\mathrm{bd}}$ and $m_o \in (0, 1)$. Let $\hat{q}_n = \frac{1}{n} \sum_{i=1}^{n} \delta_{X_i}$ be the empirical distribution of i.i.d. $X_i \sim q$. Then, under Assumption 4.1,*

$$\sqrt{n}\left(\mathrm{KL}_{\mathrm{inf}}(\hat{q}_n, m_o) - \mathrm{KL}_{\mathrm{inf}}(q, m_o)\right)$$
$$\stackrel{d}{\Rightarrow} \mathcal{N}\left(0, \sigma^2(q, m_o)\right),$$

*where*

$$\sigma^2(q, m_o) = \mathrm{Var}_q\left(\ell(\lambda^\star, X)\right) < \infty.$$

As mentioned earlier, the main difficulty in analyzing the CLT for $\mathrm{KL}_{\inf}$ statistic is that it is defined via an optimization problem, which necessitates a case-by-case treatment depending on whether the dual maximizer lies in the interior or on the boundary. We present a proof sketch of this theorem in Section 4.1, and defer the complete proof to Appendix A.1.

Next, to get the centering in the CLT for $\tau_\alpha$, we need to find its almost sure limit. To this end, we first prove a result which states that $\tau_\alpha/\log(1/\alpha)$ converges to $1/\mathrm{KL}_{\inf}(q, m_o)$ almost surely when alternate hypothesis is true. Next result formalizes this.

**Lemma 4.3.** *Fix* $q \in \mathcal{Q}^{\mathrm{bd}}$.

$$\mathbb{P}_q \left( \lim_{\alpha \to 0} \frac{\tau_\alpha}{\log(1/\alpha)} = \frac{1}{\mathrm{KL}_{\inf}(q, m_o)} \right) = 1. \quad (4)$$

We next obtain the CLT for $\tau_\alpha$ using the above two results.

**Theorem 4.4.** *Fix* $q \in \mathcal{Q}^{\mathrm{bd}}$ *and* $m_o \in (0, 1)$. *Then, under Assumption 4.1, the following holds, as* $\alpha \downarrow 0$:

$$\sqrt{\log(1/\alpha)} \left( \frac{\tau_\alpha}{\log(1/\alpha)} - \frac{1}{\mathrm{KL}_{\inf}(q, m_o)} \right)$$
$$\stackrel{d}{\Rightarrow} \mathcal{N}(0, \ \sigma_{\mathrm{bd}}^2(q, m_o)),$$

*where,*

$$\sigma_{\mathrm{bd}}^2(q, m_o) = \frac{\mathrm{Var}_q(\ell(\lambda^\star, X))}{(\mathrm{KL}_{\inf}(q, m_o))^3}.$$

A complete and rigorous proof of this theorem is deferred to the Appendix. We give the intuition and key steps of the proof now.

From Theorem 4.2, we have CLT for $\sqrt{n}(\mathrm{KL}_{\inf}(\hat{q}_n, m_o) - \mathrm{KL}_{\inf}(q, m_o))$. To prove CLT for $\tau_\alpha$, we will next develop a CLT for the same statistic at the stopping time $\tau_\alpha$. This corresponds to proving that the following converges to a Gaussian limit as $\alpha \downarrow 0$:

$$\sqrt{\tau_\alpha} \left( \mathrm{KL}_{\inf}(\hat{q}_{\tau_\alpha}, m_o) - \mathrm{KL}_{\inf}(q, m_o) \right).$$

As is standard in literature, the proof relies on Anscombe's theorem (Gut, 2009, Chapter 1). We verify Anscombe's condition in Lemma A.8. This corresponds to showing that the process is uniformly continuous in probability. However, the usual argument, which combines i.i.d. increments with Kolmogorov's maximal inequality, does not apply here, since our statistic is defined through an optimization:

$$\mathrm{KL}_{\inf}(\hat{q}_n, m_o) = \sup_{\lambda \in \left[0, \frac{1}{1-m_o}\right]} \mathbb{E}_{\hat{q}_n} \left[ \log(1 - \lambda(X - m_o)) \right],$$

when $m_o > m(\hat{q}_n)$. However, observe that,

$$\sqrt{n} \left( \mathrm{KL}_{\inf}(\hat{q}_n, m_o) - \mathrm{KL}_{\inf}(q, m_o) \right) = (T_{1,n} + T_{2,n}).$$

where $T_{1,n}$ and $T_{2,n}$ are defined later in (5) and (6), respectively. It is worth noting that $T_{2,n}$ can be handled using Kolmogorov's maximal inequality to show uniform continuity in probability, since it is a partial-sum term with i.i.d. increments. For $T_{1,n}$, we perform a Taylor expansion and exploit the first-order optimality conditions of the dual optimization problem to obtain the desired result.

Once we have a CLT for $\sqrt{\tau_\alpha}(\mathrm{KL}_{\inf}(\hat{q}_{\tau_\alpha}, m_o) - \mathrm{KL}_{\inf}(q, m_o))$, rest of the steps are standard in literature: we invert the relation via a delta-method argument to obtain a CLT for the properly rescaled stopping time.

We conclude this section with a proof sketch for Theorem 4.2. But before that, we introduce some notation. Define

$$\Phi(q, m_o) := \mathbb{E}_q \left[ \frac{1 - m_o}{1 - X} \right],$$

where by convention, we let $\Phi(q, m_o) = +\infty$ if $q(\{1\}) > 0$. Next, for $\lambda \in I_{m_o}$, $x \in [0, 1]$, and

$$g(\lambda, x) := \frac{x - m_o}{1 - \lambda(x - m_o)},$$

define

$$\Psi(\lambda, q) := \mathbb{E}_q[g(\lambda, X)],$$

which denotes the negative derivative of $\mathbb{E}_q[\ell(\lambda, X)]$ with respect to $\lambda$. Then $(\lambda^\star, q)$ satisfy exactly one of the following cases.

- **Case 1:** If $\Phi(q, m_o) < 1$, then the supremum in the dual problem is attained at the boundary, that is, $\lambda^\star = \bar{\lambda} := 1/(1 - m_o)$.

- **Case 2:** If $\Phi(q, m_o) > 1$, then the maximizer $\lambda^\star < \bar{\lambda} := 1/(1 - m_o)$ is the unique maximizer which lies in the interior of the dual feasible region, that is, it is the unique $\lambda$ that satisfies $\Psi(\lambda, q) = 0$.

- **Case 3:** If $\Phi(q, m_o) = 1$, then $\lambda^\star = \bar{\lambda} := 1/(1 - m_o)$, and $\Psi(\lambda^\star, q) = 0$.

In most of our proofs, we handle these cases separately.

### 4.1. Proof Sketch of Theorem 4.2.

Let $\lambda_n^\star := \lambda^\star(\hat{q}_n)$ be the dual maximizer. Results for the two case — $m_o > m(q)$ and $m_o < m(q)$ — follow analogously. We only provide proofs for the former.

**Case 1.** In this case, since $\lambda^\star = \bar{\lambda}$,

$$\mathrm{KL}_{\inf}(q, m_o) = \mathbb{E}_q[\ell(\bar{\lambda}, X)].$$

Therefore,

$$\sqrt{n} \left( \mathrm{KL}_{\inf}(\hat{q}_n, m_o) - \mathrm{KL}_{\inf}(q, m_o) \right) =$$

$$\sqrt{n}\left(\frac{1}{n}\sum_{i=1}^{n}\ell(\lambda_n^\star, X_i) - \mathbb{E}_q\ell(\lambda^\star, X)\right).$$

By the Strong Law of Large Numbers (SLLN),

$$\Phi(\hat{q}_n, m_o) = \frac{1}{n}\sum_{i=1}^{n}\frac{1-m_o}{1-X_i} \xrightarrow{\text{a.s.}} \Phi(q, m_o),$$

which is strictly less than 1. Thus, eventually, $(\hat{q}_n, \lambda_n^\star)$ satisfy Case 1, and hence, $\lambda_n^\star = \bar{\lambda}$ almost surely, eventually.

Finally, letting $Y_i = \ell(\lambda^\star, X_i)$ and $\bar{Y}_n := \frac{1}{n}\sum_{i=1}^{n}Y_i$, the classical CLT yields

$$\sqrt{n}(\bar{Y}_n - \mathbb{E}_q Y_1) \xRightarrow{d} \mathcal{N}(0, \sigma^2(q, m_o))$$

in this case.

**Cases 2 and 3.** Write

$$\sqrt{n}\left(\text{KL}_{\text{inf}}(\hat{q}_n, m_o) - \text{KL}_{\text{inf}}(q, m_o)\right) = T_{1,n} + T_{2,n}.$$

Where,

$$T_{1,n} = \sqrt{n}\frac{1}{n}\sum_{i=1}^{n}\left(\ell(\lambda_n^\star, X_i) - \ell(\lambda^\star, X_i)\right), \quad (5)$$

$$T_{2,n} = \sqrt{n}\left(\frac{1}{n}\sum_{i=1}^{n}\ell(\lambda^\star, X_i) - \mathbb{E}_q[\ell(\lambda^\star, X)]\right). \quad (6)$$

Recall that we have $\Phi(q, m_o) \geq 1$ for cases 2 and 3. This implies that $q(\{1\}) = 0$, so $\ell(\lambda^\star, X_i)$ is almost surely well-defined. At a high level, we will show that $T_{2,n}$ satisfies CLT and that $T_{1,n}$ is negligible (in an appropriate sense).

**Step 1: CLT for $T_{2,n}$.** Let $Y_i = \ell(\lambda^\star, X_i)$. Clearly, $(Y_i)_{i \geq 1}$ are i.i.d. with finite variance. Define $\bar{Y}_n := \frac{1}{n}\sum_{i=1}^{n}Y_i$. Then,

$$T_{2,n} = \sqrt{n}(\bar{Y}_n - \mathbb{E}_q[Y_1]) \xRightarrow{d} \mathcal{N}(0, \sigma^2(q, m_o)).$$

**Step 2: Negligibility of $T_{1,n}$.** Define

$$Q_n(\lambda) := \sum_{i=1}^{n}\left(\ell(\lambda, X_i) - \ell(\lambda^\star, X_i)\right).$$

A first-order Taylor expansion gives

$$Q_n(\lambda_n^\star) = -(\lambda_n^\star - \lambda^\star)\sum_{i=1}^{n}g(c_n, X_i),$$

where $c_n$ lies between $\lambda_n^\star$ and $\lambda^\star$. Hence,

$$T_{1,n} = -\sqrt{n}(\lambda_n^\star - \lambda^\star)A_n,$$

with

$$A_n := \frac{1}{n}\sum_{i=1}^{n}g(c_n, X_i).$$

From Lemmas A.6, and A.7,

$$\sqrt{n}(\lambda_n^\star - \lambda^\star) = O_p(1), \qquad A_n \xrightarrow{\text{a.s.}} 0,$$

so $T_{1,n} \xrightarrow{p} 0$.

Lemmas A.6 states that the dual maximizer $\lambda_n^*$ of $\text{KL}_{\text{inf}}(\hat{q}_n, m_o)$ converges almost surely to $\lambda^\star$. Further, the rate of convergence is also characterized via a CLT for $\lambda_n^\star$.

**Step 3: Conclusion.** In both the cases 2 and 3, $T_{1,n} \xrightarrow{p} 0$, while $T_{2,n} \xRightarrow{d} \mathcal{N}(0, \sigma^2(q, m_o))$. By Slutsky's theorem,

$$\sqrt{n}\left(\text{KL}_{\text{inf}}(\hat{q}_n, m_o) - \text{KL}_{\text{inf}}(q, m_o)\right) \xRightarrow{d} \mathcal{N}(0, \sigma^2(q, m_o)).$$

This completes the proof of the theorem. □

### 4.2. Application: Asymptotically Valid CI for the Stopping Time

We now leverage the stopping-time CLT to construct an asymptotically valid confidence interval for the stopping time. Since the limiting variance depends on the unknown law $q$, we estimate it consistently via a plug-in variance estimator evaluated at the stopping time, and plug it into the Gaussian limit. Notably, unlike standard simulation-based confidence intervals, which typically require many independent replications, Proposition 4.5 below provides an asymptotically valid confidence interval from the same run that produces the stopping time.

**Proposition 4.5.** *Fix $q \in \mathcal{Q}^{\text{bd}}$ and $m_o \in (0, 1)$. Define the plug-in variance estimator*

$$\hat{\sigma}_n^2 := \frac{1}{n}\sum_{i=1}^{n}\left(\ell(\lambda_n^\star, X_i) - \text{KL}_{\text{inf}}(\hat{q}_n, m_o)\right)^2,$$

*and set $\hat{v}_\alpha := \hat{\sigma}_{\tau_\alpha}^2/(\text{KL}_{\text{inf}}(\hat{q}_{\tau_\alpha}, m_o))^3$. Then, under Assumption 4.1, following holds:*

$$\hat{v}_\alpha \xrightarrow{\text{a.s.}} \sigma_{\text{bd}}^2(q, m_o).$$

*Further, for any $\gamma \in (0, 1)$, with $z_{1-\gamma/2}$ the $(1 - \gamma/2)$-quantile of $\mathcal{N}(0, 1)$, the interval*

$$\mathcal{I}_\alpha(\gamma) := \left[\frac{\tau_\alpha}{\log(1/\alpha)} \pm z_{1-\gamma/2}\sqrt{\frac{\hat{v}_\alpha}{\log(1/\alpha)}}\right]$$

*is an asymptotically valid $(1 - \gamma)$ confidence interval for $1/\text{KL}_{\text{inf}}(q, m_o)$, i.e.*

$$\lim_{\alpha\downarrow 0}\mathbb{P}\left(\frac{1}{\text{KL}_{\text{inf}}(q, m_o)} \in \mathcal{I}_\alpha(\gamma)\right) = 1 - \gamma.$$

Informally, the proposition states that $\mathcal{I}_\alpha(\gamma)$ is a confidence interval for the constant $1/\mathrm{KL}_{\inf}(q, m_o)$, which is the limit of both $\tau_\alpha / \log(1/\alpha)$ as $\alpha \to 0$ a.s. and $\mathbb{E}_q[\tau_\alpha]/\log(1/\alpha)$ as $\alpha \to 0$. Thus, the interval estimates the leading-order normalized stopping time, even though $q$ is unknown. Crucially, $\mathcal{I}_\alpha(\gamma)$ is constructed from a single sample path: the variance estimator utilized in the interval is computed along the same run that produces $\tau_\alpha$, so no independent replicates are needed.

# 5. Numerical experiments

We complement our theoretical results with three numerical studies: two synthetic simulations and one real-data experiment, examining (i) the CLT for the plug-in $\mathrm{KL}_{\inf}$ statistic, (ii) the CLT for $\tau_\alpha$, i.e., the stopping time of the $\mathrm{KL}_{\inf}$-based sequential test, and (iii) the applicability of the stopping-time CLT on a real-world dataset. All experiments consider bounded observations in $[0, 1]$.

**Experiment 1 (CLT for $\mathrm{KL}_{\inf}(\hat{q}_n, m_o)$).** We set $m_o = 0.7$ and consider two data-generating distributions on $[0, 1]$: $q \sim \mathrm{Beta}(3, 2)$ and $q \sim \mathrm{Bernoulli}(0.6)$ .

For each distribution, we generate 5000 independent i.i.d. samples of size $n$, compute $\mathrm{KL}_{\inf}(\hat{q}_n, m_o)$, and form the standardized statistic

$$\sqrt{n}\Big(\mathrm{KL}_{\inf}(\hat{q}_n, m_o) - \mathrm{KL}_{\inf}(q, m_o)\Big).$$

Figures 1 and 2 plot, for each $n$, the empirical distribution (histogram with fixed number of bins (70)) of this statistic together with the reference Gaussian density $\mathcal{N}(0, \sigma^2(q, m_o))$, illustrating convergence toward the predicted limit as $n$ increases.

It is worth noting that, in all simulations, $\mathrm{KL}_{\inf}(\hat{q}_n, m_o)$ is computed via its dual representation as a one-dimensional convex optimization problem, which can be solved efficiently using standard numerical routines.

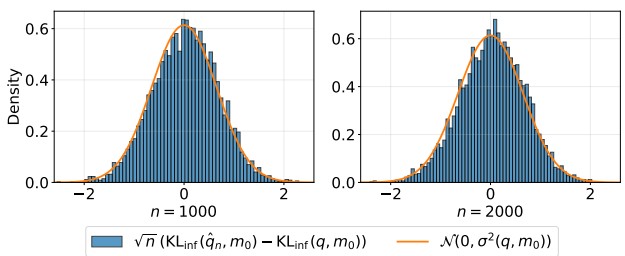

*Figure 1.* Histogram of the statistic $\sqrt{n}(\mathrm{KL}_{\inf}(\hat{q}_n, m_o) - \mathrm{KL}_{\inf}(q, m_o))$ when $q \sim \mathrm{Beta}(3, 2)$. The orange curve is the density of $\mathcal{N}(0, \sigma^2(q, m_o))$.

**Experiment 2 (CLT for the stopping time).** We now study the asymptotic normality of the stopping rule $\tau_\alpha$

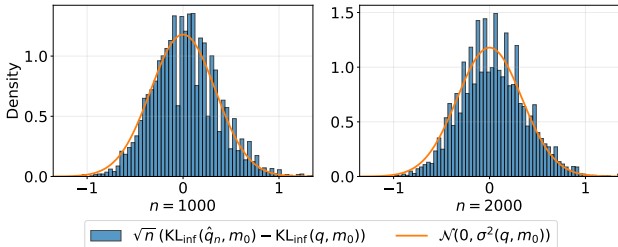

*Figure 2.* Histogram of the statistic $\sqrt{n}(\mathrm{KL}_{\inf}(\hat{q}_n, m_o) - \mathrm{KL}_{\inf}(q, m_o))$ when $q \sim \mathrm{Bernoulli}(0.6)$. The orange curve is the density of $\mathcal{N}(0, \sigma^2(q, m_o))$.

defined in (3). We set $m_o = 0.2$ and consider the data-generating distribution $q \sim \mathrm{Bernoulli}(0.6)$. We consider two values of $\alpha$: $10^{-4}, 10^{-8}$.

For each confidence level $\alpha$, we simulate 5000 independent sample paths, compute $\tau_\alpha$ along each path, and form the centered-and-scaled statistic suggested by Theorem 4.4:

$$\sqrt{\log(1/\alpha)}\left(\frac{\tau_\alpha}{\log(1/\alpha)} - \frac{1}{\mathrm{KL}_{\inf}(q, m_o)}\right).$$

The resulting empirical distribution (histogram with fixed number of bins (35)) is then compared with the corresponding limiting Gaussian distribution, $\mathcal{N}\big(0, \sigma_{\mathrm{bd}}^2(q, m_o)\big)$.

We consider two choices of the function $\beta(n, \alpha)$. The first is the theoretically-supported threshold (it is mentioned in Section 3.2) given by

$$\beta(n, \alpha) = 1 + \log\Big(\frac{2(1+n)}{\alpha}\Big). \tag{7}$$

The second is a commonly-used practical alternative (see Appendix C for more discussion on it),

$$\beta(n, \alpha) = \log(1/\alpha), \tag{8}$$

which ignores the additional $\log(1 + n)$ term and yields a flatter threshold. The results are shown in Figures 3 and 4.

The simulations reveal a clear qualitative difference across regimes. For $\alpha = 10^{-4}$ with the theoretically-supported $\beta(n, \alpha)$ given in (7), the histogram of the standardized statistic is noticeably right-skewed, indicating non-negligible finite-sample effects: many paths stop relatively early due to moderate upward fluctuations, while a smaller but non-trivial fraction require substantially longer times to cross the slowly increasing boundary, producing a long right tail. In contrast, for $\alpha = 10^{-8}$ under the practical choice (8), the empirical distribution is substantially closer to a Gaussian shape. Here, typical stopping times are much larger, placing the procedure deeper into the asymptotic regime, and the constant boundary reduces the additional distortion induced by the $\log(1 + n)$ growth. Overall, these experiments illustrate that Gaussian approximations for $\tau_\alpha$ can be sensitive

to the choice of $\beta(n, \alpha)$ and to the magnitude of $\alpha$, with the fit improving markedly when $\alpha$ is small enough that $\tau_\alpha$ is typically large.

For the last case shown in Figure 4, where $\beta(n, \alpha) = \log(1/\alpha)$, the empirical 95% confidence intervals for the simulated stopping-time distribution are $[6.0, 50.0]$ and $[22.0, 84.0]$ for $\alpha = 10^{-4}$ and $\alpha = 10^{-8}$, respectively. By comparison, the intervals obtained from a single simulation path via Proposition 4.5, after multiplying by $\log(1/\alpha)$ to put them on the stopping-time scale, are $[2.0, 41.0]$ and $[11.9, 49.0]$ for $\alpha = 10^{-4}$ and $\alpha = 10^{-8}$, respectively.

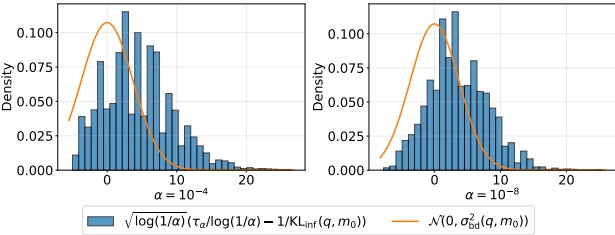

*Figure 3.* Histogram of the statistic $\sqrt{\log(1/\alpha)}(\tau_\alpha/\log(1/\alpha) - 1/\mathrm{KL}_{\mathrm{inf}}(q, m_o))$ with $\alpha = 10^{-4}$ on left and $\alpha = 10^{-8}$ on right. The orange curve is the density of $\mathcal{N}(0, \sigma_{\mathrm{bd}}^2(q, m_o))$. The choice of $\beta(n, \alpha)$ is given in (7).

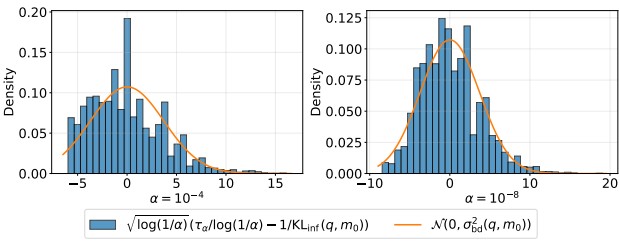

*Figure 4.* Histogram of the statistic $\sqrt{\log(1/\alpha)}(\tau_\alpha/\log(1/\alpha) - 1/\mathrm{KL}_{\mathrm{inf}}(q, m_o))$ with $\alpha = 10^{-4}$ on left and $\alpha = 10^{-8}$ on right. The orange curve is the density of $\mathcal{N}(0, \sigma_{\mathrm{bd}}^2(q, m_o))$. The choice of $\beta(n, \alpha)$ is given in (8).

**Experiment 3 (Numerical experiments on real world data).** To complement our synthetic experiments, we evaluate our sequential test on a real-world DSSAT crop-yield dataset (https://dssat.net), a physics-based crop-growth model widely used in agronomy research. In DSSAT, the yields are bounded after normalization and exhibit a nonparametric distributional shape, making them a natural testbed.

We set the null hypothesis value to $m_o = 0.5$ and run our sequential test at significance level $\alpha = 10^{-4}$. To approximate the stopping-time distribution, we generate 3,000 independent bootstrap paths by resampling with replacement from the data pool.

Since the underlying outcome distribution $q$ is unknown and only an empirical sample from $q$ is available, the quantities

$1/\mathrm{KL}_{\mathrm{inf}}(q, m_o)$ and $\sigma_{\mathrm{bd}}^2$ appearing in the asymptotic CLT cannot be evaluated analytically. We therefore estimate these quantities using the empirical distribution $\hat{q}$ induced by the observed data. In particular, $\sigma_{\mathrm{bd}}^2$ is estimated by plugging in $\hat{q}$, instead of $q$, in the expression for $\sigma_{\mathrm{bd}}^2$ defined in Theorem 4.4; the resulting estimate is denoted by $\hat{\sigma}_{\mathrm{bd}}^2$.

We find that the empirical distribution of $\tau_\alpha/\log(1/\alpha)$ closely matches the $\mathcal{N}(1/\mathrm{KL}_{\mathrm{inf}}(\hat{q}, m_o), \hat{\sigma}_{\mathrm{bd}}^2/\log(1/\alpha))$ overlay, providing additional empirical support for our asymptotic CLT result in a realistic, non-synthetic setting, as shown in Figure 5.

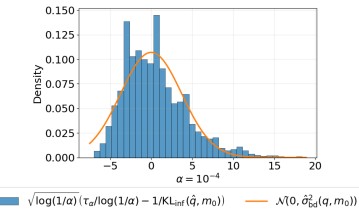

*Figure 5.* Histogram of the statistic $\sqrt{\log(1/\alpha)}(\tau_\alpha/\log(1/\alpha) - 1/\mathrm{KL}_{\mathrm{inf}}(\hat{q}, m_o))$. The orange curve is the density of $\mathcal{N}(0, \hat{\sigma}_{\mathrm{bd}}^2(q, m_o))$. The choice of $\beta(n, \alpha)$ is given in (8).

## 6. Discussion and future work

This paper develops the first stopping-time CLT for the sequential mean testing problem under a nonparametric model with bounded support. The key technical ingredient is Theorem 4.2, which establishes a CLT for $\mathrm{KL}_{\mathrm{inf}}(\hat{q}_n, m_o)$. We then leverage this result to derive a CLT for the associated stopping time in the sequential test. A limitation of our current approach is that the proof of Theorem 4.2 relies on the bounded-support structure underlying the definition and analysis of $\mathrm{KL}_{\mathrm{inf}}(\hat{q}_n, m_o)$. Extending these distributional results to broader nonparametric families, in particular, unbounded or heavy-tailed distributions, appears to require new ideas, and we leave this direction for future work.

The CLT for $\mathrm{KL}_{\mathrm{inf}}(\hat{q}_n, m_o)$ yields an asymptotically normal approximation that can be inverted to construct an asymptotically valid confidence interval for quantities defined through $\mathrm{KL}_{\mathrm{inf}}$. This may also be useful in sequential decision-making problems such as multi-armed bandits, where many algorithms employ $\mathrm{KL}_{\mathrm{inf}}$-based confidence bounds (Agrawal et al., 2021a). Our results suggest a potential route to designing bandit procedures with asymptotically calibrated confidence guarantees.

Beyond confidence calibration, it is also natural to ask whether finer distributional characterizations of $\mathrm{KL}_{\mathrm{inf}}$-based confidence processes can shed light on the tail behavior of $\mathrm{KL}_{\mathrm{inf}}$-based learning algorithms. In particular, it will be interesting to understand whether such characterizations can provide insight into the heavy regret tails recently

observed for certain asymptotically optimal bandit algorithms (Panda & Agrawal, 2026; Fan & Glynn, 2025). While Theorem 4.2 characterizes typical Gaussian fluctuations of the empirical $\mathrm{KL}_{\inf}$ statistic, the regret-tail phenomenon is driven by atypical trajectories and therefore appears to require moderate- or large-deviation analyses. Developing such results for the $\mathrm{KL}_{\inf}$ process, and understanding their implications for regret-tail behavior in multi-armed bandits, is an interesting direction for future work.

Finally, we focus on a composite vs. composite hypothesis testing problem. In this setting, beyond the $\alpha \to 0$ regime studied in this paper, the other natural asymptotic regime is $\mathrm{KL}_{\inf}(q, m_o) \to 0$, for a fixed $\alpha$. This can be achieved by considering a sequence of alternative data-generating distributions, whose mean approaches $m_o$. In this regime, the expected stopping time has been recently shown to grow as $\Omega\left(\mathrm{KL}_{\inf}^{-1} \log \log \mathrm{KL}_{\inf}^{-1}\right)$, a rate reminiscent of the law of the iterated logarithm (see Agrawal & Ramdas (2025) for more details). The reference also shows some tests achieving this rate. However, even the first-order asymptotics in this regime are not completely understood in the nonparametric setting. We therefore leave this as an interesting direction for future work.

## Impact Statement

This paper presents work whose goal is to advance the field of Machine Learning. There are many potential societal consequences of our work, none which we feel must be specifically highlighted here.

## Acknowledgements

The authors thank the anonymous reviewers for their comments and suggestions, which helped improve the presentation of this paper. SA acknowledges support from the Pratiksha Trust, Bangalore, through the Young Investigator Award, and from ANRF through grant ANRF/ECRG/2025/000560/ENS.

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

# A. Proofs for Main Results

In this appendix, we will provide detailed proofs of the results from Section 4, and also present and prove supporting lemmas. Towards this, first recall that

$$\Phi(q, m_o) := \mathbb{E}_q\left[\frac{1 - m_o}{1 - X}\right],$$

where by convention, we let $\Phi(q, m_o) = +\infty$ if $q(\{1\}) > 0$. Further,

$$\Psi(\lambda, q) = \mathbb{E}_q\left[g(\lambda, X)\right] := \mathbb{E}_q\left[\frac{X - m_o}{1 - \lambda(X - m_o)}\right],$$

which denotes the negative derivative of $\mathbb{E}_q[\ell(\lambda, X)]$ with respect to $\lambda$. Then, one of the following three cases occurs.

- **Case 1:** If $\Phi(q, m_o) < 1$, then the supremum in the dual problem is attained at the boundry, that is, $\lambda^\star = \bar{\lambda} = \frac{1}{1 - m_o}$.

- **Case 2:** If $\Phi(q, m_o) > 1$, then the maximizer $\lambda^\star < \bar{\lambda} := \frac{1}{1 - m_o}$ is the unique maximizer which lies in the interior of the dual feasible region, that is, it is the unique $\lambda$ that satisfies $\Psi(\lambda, q) = 0$.

- **Case 3:** If $\Phi(q, m_o) = 1$, then $\lambda^\star = \bar{\lambda} = \frac{1}{1 - m_o}$, and $\Psi(\lambda^\star, q) = 0$.

*Remark* A.1. Under Case 2, the choice of $q$ and $m_o$ can lead to $\lambda^* = 0$. In this scenario, $\mathrm{KL}_{\inf}(q, m_o) = 0$, which holds if and only if $m(q) = m_o$ (see, for example, Jourdan et al. (2022, Appendix F)) . However, all of the results in this paper assume that $q \in \mathcal{Q}^{\mathrm{bd}}$, which implies that $m(q) \neq m_o$. Hence, without loss of generality, we assume that under this case, when $\lambda^* < \frac{1}{1 - m_o}$, it also holds that $\lambda^* > 0$.

Our proofs often treat these cases separately. Finally, before presenting our proofs, we introduce certain notation which will be used in this Appendix.

## Notation

- Let $g'(c, x) = \frac{\partial g(\lambda, x)}{\partial \lambda}\big|_{\lambda = c}$ and $\Psi'(c, q) := \frac{\partial}{\partial \lambda}\Psi(\lambda, q)\big|_{\lambda = c}$.

- Convergence notation: $\xrightarrow{\text{a.s.}}$ almost surely, $\xrightarrow{P}$ in probability, $\xrightarrow{d}$ in distribution.

- Stochastic order notation: $X_n = O_p(1)$ means $\{X_n\}$ is bounded in probability, i.e., for every $\epsilon > 0$ there exists $M < \infty$ such that $\sup_n \Pr(|X_n| > M) < \epsilon$. More generally, $X_n = O_p(a_n)$ means $X_n/a_n = O_p(1)$. We write $X_n = o_p(1)$ if $X_n \xrightarrow{P} 0$, and $X_n = o_p(a_n)$ if $X_n/a_n \xrightarrow{P} 0$.

## A.1. Proof of Theorem 4.2.

As mentioned earlier, we prove the theorem for the case when $m_o > m(q)$. The other case (when $m_o < m(q)$) follows analogously.

*Proof.* We begin by recalling from Lemma A.3 that $\sigma^2(q, m_o) := \mathrm{Var}_q(\ell(\lambda^\star, X))$ is finite. This will be used in applying classical CLT at appropriate steps, later in the proof. We now prove the result in the three cases (introduced at the beginning of Appendix A) separately.

**Case 1.** In this case, since $\lambda^\star = \bar{\lambda}$, $\mathrm{KL}_{\inf}(q, m_o) = \mathbb{E}_q[\ell(\bar{\lambda}, X)]$, and hence,

$$\sqrt{n}\Big(\mathrm{KL}_{\inf}(\hat{q}_n, m_o) - \mathrm{KL}_{\inf}(q, m_o)\Big) = \sqrt{n}\left[\frac{1}{n}\sum_{i=1}^n \big(\ell(\lambda_n^\star, X_i) - \mathbb{E}_q[\ell(\bar{\lambda}, X)]\big)\right].$$

Further, SLLN gives

$$\frac{1}{n}\sum_{i=1}^n \frac{1 - m_o}{1 - X_i} \xrightarrow{\text{a.s.}} \Phi(q, m_o) < 1.$$

Hence, on each sample path $\omega$, there exists $N(\omega)$ such that for all $n \geq N(\omega)$,

$$\frac{1}{n} \sum_{i=1}^{n} \frac{1 - m_o}{1 - X_i} = \Phi(\hat{q}_n, m_o) < 1,$$

which implies $\lambda_n^\star = \bar{\lambda}$ for all $n \geq N(\omega)$ on each sample path $\omega$. Therefore, on each path $\omega$, for such large $n$,

$$\sqrt{n}\Big(\mathrm{KL}_{\mathrm{inf}}(\hat{q}_n, m_o) - \mathrm{KL}_{\mathrm{inf}}(q, m_o)\Big) = \sqrt{n}\left[\frac{1}{n} \sum_{i=1}^{n} \big(\ell(\bar{\lambda}, X_i) - \mathbb{E}_q[\ell(\bar{\lambda}, X)]\big)\right].$$

Let $Y_i = \ell(\bar{\lambda}, X_i)$ and $\bar{Y}_n = \frac{1}{n} \sum_{i=1}^{n} Y_i$. Then $(Y_i)$ are i.i.d. with mean $\mathbb{E}_q[Y_1] = \mathbb{E}_q[\ell(\bar{\lambda}, X)] = \mathrm{KL}_{\mathrm{inf}}(q, m_o)$ and variance $\sigma^2(q, m_o) = \mathrm{Var}_q(\ell(\bar{\lambda}, X))$, which is finite. Hence by the classical CLT for i.i.d. random variables,

$$\sqrt{n}\Big(\mathrm{KL}_{\mathrm{inf}}(\hat{q}_n, m_o) - \mathrm{KL}_{\mathrm{inf}}(q, m_o)\Big) = \sqrt{n}\big(\bar{Y}_n - \mathbb{E}_q[Y_1]\big) \overset{d}{\Rightarrow} \mathcal{N}\big(0, \sigma^2(q, m_o)\big).$$

This completes the proof for this case.

**Cases 2 and 3.** Again, recall that $\mathrm{KL}_{\mathrm{inf}}(\cdot, m_o) = \mathbb{E}.[\ell(\lambda^\star(\cdot), X)]$. Thus,

$$\sqrt{n}\Big(\mathrm{KL}_{\mathrm{inf}}(\hat{q}_n, m_o) - \mathrm{KL}_{\mathrm{inf}}(q, m_o)\Big)$$
$$= \sqrt{n}\left[\frac{1}{n} \sum_{i=1}^{n} \big(\ell(\lambda_n^\star, X_i) - \ell(\lambda^\star, X_i)\big)\right] + \sqrt{n}\left(\frac{1}{n} \sum_{i=1}^{n} \ell(\lambda^\star, X_i) - \mathbb{E}_q[\ell(\lambda^\star, X)]\right)$$
$$=: T_{1,n} + T_{2,n}.$$

*Remark* A.2. In Case 3 (when $\lambda^\star = \bar{\lambda}$), $\ell(\lambda^\star, X_i) := \log(1 - \lambda^\star(X_i - m_o))$ is undefined for $X_i = 1$. However, we have $\Phi(q, m_o) = 1$, which implies that $q(\{1\}) = 0$. Hence, on a set of measure one, $X_i < 1$ in this case. Therefore, without loss of generality, for Case 3, we will assume that $X_i < 1$ for all $i \in \mathbb{N}$.

We will now handle the two terms $T_{2,n}$ and $T_{1,n}$ separately.

**Step 1: Analyzing $T_{2,n}$.** Let $Y_i = \ell(\lambda^\star, X_i)$ and $\bar{Y}_n = \frac{1}{n} \sum_{i=1}^{n} Y_i$. Then $(Y_i)$ are i.i.d. with mean $\mathbb{E}_q[\ell(\lambda^\star, X)]$ and variance $\sigma^2(q, m_o) < \infty$, then the classical CLT holds and gives

$$T_{2,n} = \sqrt{n}\big(\bar{Y}_n - \mathbb{E}_q[\ell(\lambda^\star, X)]\big) \overset{d}{\Rightarrow} \mathcal{N}\big(0, \sigma^2(q, m_o)\big).$$

**Step 2: Analyzing $T_{1,n}$.** For fixed $X_1, \ldots, X_n$, consider the following function:

$$Q_n : \lambda \to \sum_{i=1}^{n} \big(\ell(\lambda, X_i) - \ell(\lambda^\star, X_i)\big).$$

The function $Q_n$ is continuously differentiable in a neighborhood containing $\lambda^\star$. This follows from the definition of $\ell(\cdot, X)$ in Case 3, since $X_i \in [0, 1)$ almost surely (hence $\ell(X_i, \lambda) > 0$ and infinitely differentiable for all $\lambda \in [0, \frac{1}{1-m_o}]$). In Case 2, we have from Lemma A.6 that on each sample path (for all realizations of $X_i \in [0, 1]$), the dual optimizer $\lambda_n^\star$ lies in the interior for sufficiently large $n$. In particular, $\lambda_n^\star \in [\lambda^\star - \eta, \lambda^\star + \eta]$ for sufficiently large $n$, where $\eta > 0$ is chosen such that $\lambda^\star + \eta < \frac{1}{1-m_o}$ and $\lambda^\star - \eta > 0$. Hence, $\ell(\lambda, X_i) > 0$ for all $i$ and for all $\lambda$ in this neighborhood containing $\lambda^\star$.

Hence, using Taylor expansion of $Q_n(\cdot)$ around $\lambda^\star$ for large $n$ yields

$$Q_n(\lambda_n^\star) = \underbrace{Q_n(\lambda^\star)}_{=0} + (\lambda_n^\star - \lambda^\star) \frac{\partial Q_n(\lambda)}{\partial \lambda}\bigg|_{\lambda = c(X_1, X_2, \ldots X_n, \lambda_n^\star)},$$

for some $c(X_1, \ldots, X_n, \lambda_n^\star)$. For ease of notation, we let $c_n := c(X_1, X_2, \ldots X_n, \lambda_n^\star)$.

Now, observe that $T_{1,n} = Q_n(\lambda_n^\star)/\sqrt{n}$. Hence,

$$T_{1,n} = -\sqrt{n}(\lambda_n^\star - \lambda^\star) \cdot A_n, \qquad A_n := \frac{1}{n} \sum_{i=1}^{n} g(c_n, X_i),$$

for some (possibly random) $c_n$ between $\lambda_n^\star$ and $\lambda^\star$. Now, from Lemma A.6, we have the following

$$\lambda_n^\star \xrightarrow{\text{a.s.}} \lambda^\star \quad \text{and} \quad \sqrt{n}\,(\lambda^\star - \lambda_n^\star) = O_p(1),$$

for cases 2 and 3, respectively. Further, Lemma A.7 gives that $A_n \to 0$ almost surely. Hence, it follows that $T_{1,n} \xrightarrow{p} 0$ under Case 2 and Case 3.

**Step 3: Conclusion.** In both the cases, Case 2 and Case 3, $T_{1,n} \xrightarrow{p} 0$, while $T_{2,n} \xrightarrow{d} \mathcal{N}(0, \sigma^2(q, m_o))$. Thus, by Slutsky's theorem,

$$\sqrt{n}\Big(\mathrm{KL}_{\mathrm{inf}}(\hat{q}_n, m_o) - \mathrm{KL}_{\mathrm{inf}}(q, m_o)\Big) \xrightarrow{d} \mathcal{N}\big(0, \sigma^2(q, m_o)\big).$$

This completes the proof. □

## A.2. Proof of Lemma 4.3

We prove the lemma for the case $m_o > m(q)$; the proof for the case $m_o < m(q)$ follows analogously.

First we show that

$$\lim_{\alpha \to 0} \tau_\alpha = \infty \qquad \text{almost surely.} \tag{9}$$

Recall the stopping time

$$\tau_\alpha = \inf\Big\{ n \geq 1 : \ n\,\mathrm{KL}_{\mathrm{inf}}(\hat{q}_n, m_o) \ \geq \ \beta(n, \alpha)\Big\}.$$

Using Honda & Takemura (2010, Lemma 14), we have the uniform bound

$$\mathrm{KL}_{\mathrm{inf}}(\hat{q}_n, m_o) \ \leq \ \frac{1}{1 - m_o}, \qquad \text{for all } n \geq 1. \tag{10}$$

Consequently,

$$n\,\mathrm{KL}_{\mathrm{inf}}(\hat{q}_n, m_o) \ \leq \ \frac{n}{1 - m_o}, \qquad \text{for all } n \geq 1. \tag{11}$$

Fix an arbitrary integer $N \geq 1$. For all $1 \leq n \leq N$,

$$n\,\mathrm{KL}_{\mathrm{inf}}(\hat{q}_n, m_o) \ \leq \ \frac{N}{1 - m_o}.$$

On the other hand, since $\beta(n, \alpha) = O(\log(n/\alpha))$, we have

$$\lim_{\alpha \downarrow 0} \beta(n, \alpha) \ \to \ \infty \qquad \text{for each fixed } n.$$

Hence, there exists $\alpha_N \in (0, 1)$ such that

$$\min_{1 \leq n \leq N} \beta(n, \alpha) \ > \ \frac{N}{1 - m_o}, \qquad \text{for all } \alpha \in (0, \alpha_N). \tag{12}$$

Combining (11) and (12), we obtain that for all $\alpha \in (0, \alpha_N)$ and all $1 \leq n \leq N$,

$$n\,\mathrm{KL}_{\mathrm{inf}}(\hat{q}_n, m_o) \ < \ \beta(n, \alpha).$$

Therefore, using the definition of $\tau_\alpha$, it follows that

$$\tau_\alpha > N \qquad \text{for all } \alpha \in (0, \alpha_N).$$

Since $N \geq 1$ was arbitrary, it follows that

$$\lim_{\alpha \to 0} \tau_\alpha = \infty \quad \text{almost surely.}$$

Now we prove the statement of the lemma. Since $q \in \mathcal{Q}^{\mathrm{bd}}$, and the fact that this test has power one, following holds for any fixed $\alpha \in (0, 1)$:

$$\mathbb{P}_q(\tau_\alpha < \infty) = 1 \text{ almost surely.}$$

Using the above fact and the definition of $\tau_\alpha$, we get,

$$\frac{\beta(\tau_\alpha - 1, \alpha)}{\log(1/\alpha)\,\mathrm{KL}_{\mathrm{inf}}(\hat{q}_{\tau_\alpha-1}, m_o)} + \frac{1}{\log(1/\alpha)} \geq \frac{\tau_\alpha}{\log(1/\alpha)} \geq \frac{\beta(\tau_\alpha, \alpha)}{\log(1/\alpha)\,\mathrm{KL}_{\mathrm{inf}}(\hat{q}_{\tau_\alpha}, m_o)}. \tag{13}$$

First, we show that following claim holds:

*Claim :* $\tau_\alpha = O(\log(1/\alpha))$ *almost surely.*

Proof of the claim. Fix $\epsilon \in \big(0, \mathrm{KL}_{\mathrm{inf}}(q, m_o)\big)$ and define the event

$$\Omega_\epsilon := \Big\{\exists\, N_\epsilon(\omega) \text{ s.t. } \mathrm{KL}_{\mathrm{inf}}(\hat{q}_n, m_o) \geq \mathrm{KL}_{\mathrm{inf}}(q, m_o) - \epsilon, \ \forall n \geq N_\epsilon(\omega)\Big\}.$$

Further, fix $\delta \in \big(0, \mathrm{KL}_{\mathrm{inf}}(q, m_o) - \epsilon\big)$. Since $\log x = o(x)$ as $x \to \infty$, there exists $M_\delta < \infty$ such that

$$\log x \leq \delta x, \qquad \forall\, x \geq M_\delta. \tag{14}$$

Using Lemma A.5, we get $\mathrm{KL}_{\mathrm{inf}}(\hat{q}_n, m_o) \xrightarrow{a.s.} \mathrm{KL}_{\mathrm{inf}}(q, m_o)$. Hence, it follows that, we have $\mathbb{P}_q(\Omega_\epsilon) = 1$.

Fix $\omega \in \Omega_\epsilon$. Since $\tau_\alpha(\omega) \to \infty$ as $\alpha \downarrow 0$ (see (9)), there exists $\alpha_0(\omega) \in (0, 1)$ such that $\tau_\alpha(\omega) - 1 \geq \max\{N_\epsilon(\omega), M_\delta - 1\}$ for all $\alpha \in (0, \alpha_0(\omega))$. Using the definition of $\tau_\alpha$, we get,

$$\big(\tau_\alpha(\omega) - 1\big)\,\mathrm{KL}_{\mathrm{inf}}\big(\hat{q}_{\tau_\alpha(\omega)-1}, m_o\big) \ < \ \beta(\tau_\alpha(\omega) - 1, \alpha) \ = \ 1 + \log\Big(\frac{2\tau_\alpha(\omega)}{\alpha}\Big).$$

On $\Omega_\epsilon$ and for $\alpha < \alpha_0(\omega)$ we also have $\mathrm{KL}_{\mathrm{inf}}(\hat{q}_{\tau_\alpha(\omega)-1}, m_o) \geq \mathrm{KL}_{\mathrm{inf}}(q, m_o) - \epsilon$, hence

$$\big(\tau_\alpha(\omega) - 1\big)\big(\mathrm{KL}_{\mathrm{inf}}(q, m_o) - \epsilon\big) \ < \ 1 + \log\Big(\frac{2}{\alpha}\Big) + \log\big(\tau_\alpha(\omega)\big). \tag{15}$$

Using (14) and definition of $\alpha_0(\omega)$, we get,

$$\big(\tau_\alpha(\omega) - 1\big)\big(\mathrm{KL}_{\mathrm{inf}}(q, m_o) - \epsilon\big) \ < \ 1 + \delta + \log\Big(\frac{2}{\alpha}\Big) + \delta\big(\tau_\alpha(\omega) - 1\big).$$

Rearranging gives

$$\big(\mathrm{KL}_{\mathrm{inf}}(q, m_o) - \epsilon - \delta\big)\big(\tau_\alpha(\omega) - 1\big) \ < \ 1 + \delta + \log\Big(\frac{2}{\alpha}\Big).$$

Choose $\delta := \frac{1}{2}\big(\mathrm{KL}_{\mathrm{inf}}(q, m_o) - \epsilon\big)$, so that $\mathrm{KL}_{\mathrm{inf}}(q, m_o) - \epsilon - \delta = \frac{1}{2}\big(\mathrm{KL}_{\mathrm{inf}}(q, m_o) - \epsilon\big)$. Then, for all $\alpha \in (0, \alpha_0(\omega))$ (depending on $\omega$),

$$\tau_\alpha(\omega) \ \leq \ 1 + \frac{2}{\mathrm{KL}_{\mathrm{inf}}(q, m_o) - \epsilon}\Big(1 + \delta + \log\Big(\frac{2}{\alpha}\Big)\Big).$$

Since $1 + \delta$ is a finite constant and $\log(2/\alpha) = \log(1/\alpha) + \log 2$, this proves $\tau_\alpha = O(\log(1/\alpha))$ almost surely. Using the claim, (9), (13) and the fact that $\mathrm{KL}_{\mathrm{inf}}(\hat{q}_n, m_o) \to \mathrm{KL}_{\mathrm{inf}}(q, m_o)$ almost surely (see Lemma A.5), we get the desired result. This completes the proof.

$\square$

### A.3. Proof of Theorem 4.4

Fix $q \in \mathcal{Q}^{\mathrm{bd}}$. Recall that

$$\tau_\alpha = \inf\left\{ n : \ \mathrm{KL}_{\inf}(\hat{q}_n, m_o) \geq \frac{\beta(n, \alpha)}{n} \right\}.$$

By definition of $\tau_\alpha$,

$$\mathrm{KL}_{\inf}(\hat{q}_{\tau_\alpha}, m_o) \geq \frac{\beta(\tau_\alpha, \alpha)}{\tau_\alpha}, \qquad \mathrm{KL}_{\inf}(\hat{q}_{\tau_\alpha - 1}, m_o) < \frac{\beta(\tau_\alpha - 1, \alpha)}{\tau_\alpha - 1}. \tag{16}$$

Equivalently,

$$\frac{\beta(\tau_\alpha, \alpha)}{\mathrm{KL}_{\inf}(\hat{q}_{\tau_\alpha}, m_o)} \leq \tau_\alpha < 1 + \frac{\beta(\tau_\alpha - 1, \alpha)}{\mathrm{KL}_{\inf}(\hat{q}_{\tau_\alpha - 1}, m_o)}. \tag{17}$$

By Theorem 4.2, we have

$$\sqrt{n}\Big(\mathrm{KL}_{\inf}(\hat{q}_n, m_o) - \mathrm{KL}_{\inf}(q, m_o)\Big) \overset{d}{\Rightarrow} \mathcal{N}(0, \sigma^2(q, m_o)). \tag{18}$$

Let $b_\alpha = \log(1/\alpha)$. Since $\tau_\alpha / b_\alpha \to 1/\mathrm{KL}_{\inf}(q, m_o)$ almost surely by Lemma 4.3, and using Lemma A.8, we get that Anscombe's condition is satisfied and hence it follows that, using Anscombe's theorem, we have,

$$\sqrt{\tau_\alpha}\Big(\mathrm{KL}_{\inf}(\hat{q}_{\tau_\alpha}, m_o) - \mathrm{KL}_{\inf}(q, m_o)\Big) \overset{d}{\Rightarrow} \mathcal{N}(0, \sigma^2(q, m_o)). \tag{19}$$

Using a similar argument as above for $\tau_\alpha - 1$ instead of $\tau_\alpha$, we get,

$$\sqrt{\tau_\alpha - 1}\Big(\mathrm{KL}_{\inf}(\hat{q}_{\tau_\alpha - 1}, m_o) - \mathrm{KL}_{\inf}(q, m_o)\Big) \overset{d}{\Rightarrow} \mathcal{N}(0, \sigma^2(q, m_o)). \tag{20}$$

It is worth noting that the CLT for a test statistic evaluated at a random stopping time does not, in general, imply the CLT for it evaluated at a finite number of steps before that stopping time. However, in our setting, this follows trivially by applying Anscombe's theorem to both $\tau_\alpha$ and $\tau_\alpha - 1$.

Using Delta method, we get

$$\sqrt{\tau_\alpha}\left( \frac{1}{\mathrm{KL}_{\inf}(\hat{q}_{\tau_\alpha}, m_o)} - \frac{1}{\mathrm{KL}_{\inf}(q, m_o)} \right) \overset{d}{\Rightarrow} \mathcal{N}\left(0, \frac{\sigma^2(q, m_o)}{\mathrm{KL}_{\inf}(q, m_o)^4}\right).$$

$$\sqrt{\tau_\alpha - 1}\left( \frac{1}{\mathrm{KL}_{\inf}(\hat{q}_{\tau_\alpha - 1}, m_o)} - \frac{1}{\mathrm{KL}_{\inf}(q, m_o)} \right) \overset{d}{\Rightarrow} \mathcal{N}\left(0, \frac{\sigma^2(q, m_o)}{\mathrm{KL}_{\inf}(q, m_o)^4}\right).$$

Also, by Lemma 4.3,

$$\sqrt{\frac{b_\alpha}{\tau_\alpha}} \overset{p}{\to} \sqrt{\mathrm{KL}_{\inf}(q, m_o)}.$$

Therefore, by Slutsky's theorem, we get,

$$\sqrt{b_\alpha}\left( \frac{1}{\mathrm{KL}_{\inf}(\hat{q}_{\tau_\alpha}, m_o)} - \frac{1}{\mathrm{KL}_{\inf}(q, m_o)} \right) \overset{d}{\Rightarrow} \mathcal{N}\left(0, \frac{\sigma^2(q, m_o)}{\mathrm{KL}_{\inf}(q, m_o)^3}\right),$$

$$\sqrt{b_\alpha}\left( \frac{1}{\mathrm{KL}_{\inf}(\hat{q}_{\tau_\alpha - 1}, m_o)} - \frac{1}{\mathrm{KL}_{\inf}(q, m_o)} \right) \overset{d}{\Rightarrow} \mathcal{N}\left(0, \frac{\sigma^2(q, m_o)}{\mathrm{KL}_{\inf}(q, m_o)^3}\right). \tag{21}$$

It follows that (17) can be re-written as,

$$\sqrt{b_\alpha}\frac{\beta(\tau_\alpha, \alpha)}{b_\alpha}\left( \frac{1}{\mathrm{KL}_{\inf}(\hat{q}_{\tau_\alpha}, m_o)} - \frac{1}{\mathrm{KL}_{\inf}(q, m_o)} \right) \leq \sqrt{b_\alpha}\left( \frac{\tau_\alpha}{b_\alpha} - \frac{1}{\mathrm{KL}_{\inf}(q, m_o)} \right)$$

$$< \frac{1}{\sqrt{b_\alpha}} + \sqrt{b_\alpha}\frac{\beta(\tau_\alpha - 1, \alpha)}{b_\alpha}\left( \frac{1}{\mathrm{KL}_{\inf}(\hat{q}_{\tau_\alpha - 1}, m_o)} - \frac{1}{\mathrm{KL}_{\inf}(q, m_o)} \right).$$

Using (21), the definition of $\beta(n, \alpha)$ and Lemma 4.3, we get the desired result.

$\square$

## A.4. Proof of Proposition 4.5

We first prove the proposition for the case $m_o > m(q)$ (the case $m_o < m(q)$ uses the $\mathrm{KL}_{\mathrm{inf}}^-$ formula analogously).

Let

$$L := \mathrm{KL}_{\mathrm{inf}}(q, m_o) = \sup_{\lambda \in [0, \bar{\lambda}]} \mathbb{E}_q\big[\ell(\lambda, X)\big],$$

We use the identity

$$\hat{\sigma}_n^2 = \frac{1}{n} \sum_{i=1}^n \ell(\lambda_n^\star, X_i)^2 - \left( \frac{1}{n} \sum_{i=1}^n \ell(\lambda_n^\star, X_i) \right)^2. \tag{22}$$

Using Lemma A.5, we get

$$\hat{\sigma}_n^2 \xrightarrow{\text{a.s.}} \mathbb{E}_q[\ell(\lambda^\star, X)^2] - \left( \mathbb{E}_q[\ell(\lambda^\star, X)] \right)^2 = \mathrm{Var}_q(\ell(\lambda^\star, X)), \text{ and}$$

$$\mathrm{KL}_{\mathrm{inf}}(\hat{q}_n, m_o) \xrightarrow{\text{a.s.}} \mathrm{KL}_{\mathrm{inf}}(q, m_o).$$

Hence, by the continuous mapping theorem,

$$\frac{\hat{\sigma}_n^2}{(\mathrm{KL}_{\mathrm{inf}}(\hat{q}_n, m_o))^3} \xrightarrow{\text{a.s.}} \frac{\mathrm{Var}_q(\ell(\lambda^\star, X))}{L^3} = \frac{\mathrm{Var}_q(\ell(\lambda^\star, X))}{\mathrm{KL}_{\mathrm{inf}}(q, m_o)^3}. \tag{23}$$

Using Lemma 4.3, we also know that $\lim_{\alpha \to 0} \tau_\alpha \to \infty$ almost surely. Combining this with (23), we get the desired result. This completes the proof.

$\square$

## A.5. Supporting Lemmas

**Lemma A.3.** *Let* $q \in \mathcal{Q}^{\mathrm{bd}}$. *For* $k \in \mathbb{N}$, $\mathbb{E}_q[|\ell(\lambda^\star, X)|^k] < \infty$.

*Proof.* We prove the lemma for the case $m_o > m(q)$; the proof for the case $m_o < m(q)$ follows analogously.

To show this, we show that the moment-generating function for the random variable $\ell(\lambda^\star, X)$, when $X \sim q$, is finite in a neighborhood containing $0$, and hence, all its moments are finite. In particular, we show the following for both $\theta = 1$ and $\theta = -1$,

$$\mathbb{E}_q[e^{\theta \ell(\lambda^\star, X)}] = \mathbb{E}_q[e^{\theta \log(1 - \lambda^\star(X - m_o))}] < \infty.$$

For $\theta = 1$, we have

$$\mathbb{E}_q[e^{\theta \log(1 - \lambda^\star(X - m_o))}] = (1 - \lambda^\star(m(q) - m_o)).$$

Since $m_o > m(q)$ and $\lambda^\star \in [0, \frac{1}{1 - m_o}]$, the term on the right hand side in the above equation is finite.

Finally, for $\theta = -1$, we have

$$\mathbb{E}_q[e^{\theta \log(1 - \lambda^\star(X - m_o))}] = \mathbb{E}_q \left[ \frac{1}{1 - \lambda^\star(X - m_o)} \right] \leq 1,$$

where the last inequality follows since the middle expression in the set of inequalities above corresponds to the mass that the primal optimizer in $\mathrm{KL}_{\mathrm{inf}}(q, m_o)$ puts on the support of distribution $q$ (see (Agrawal, 2023)). $\square$

**Lemma A.4.** *Let* $\mathcal{R}$ *be either of the following sets:*

$$\mathcal{R} = I_\epsilon \times [0, 1], \qquad I_\epsilon := [\lambda^\star - \epsilon, \lambda^\star + \epsilon] \subset (0, \bar{\lambda}), \quad \lambda^\star \in (0, \bar{\lambda}),$$

*for some* $\epsilon > 0$, *or*

$$\mathcal{R} = [0, \bar{\lambda}] \times [0, 1 - \kappa], \qquad \kappa \in (0, 1), \quad 1 - \kappa > m_o.$$

*Then* $\ell(\lambda, x)$, $g(\lambda, x)$, *and* $g'(\lambda, x)$ *are uniformly bounded on* $\mathcal{R}$. *Moreover, they are uniformly Lipschitz in* $\lambda$ *on* $\mathcal{R}$.

*Proof.* First suppose $\mathcal{R} = I_\epsilon \times [0, 1]$, where $\lambda^\star \in (0, \bar{\lambda})$ and $I_\epsilon = [\lambda^\star - \epsilon, \lambda^\star + \epsilon] \subset (0, \bar{\lambda})$. Let,

$$m_\epsilon := 1 - (\lambda^\star + \epsilon)(1 - m_o) > 0.$$

For all $(\lambda, x) \in \mathcal{R}$,

$$1 - \lambda(x - m_o) \geq \begin{cases} 1, & x \leq m_o, \\ m_\epsilon, & x \geq m_o. \end{cases}$$

Hence $\ell$ is uniformly bounded on $\mathcal{R}$. Moreover,

$$|g(\lambda, x)| \leq B_\epsilon := \max\left\{ m_o, \frac{1 - m_o}{m_\epsilon} \right\} < \infty,$$

and

$$0 \leq g'(\lambda, x) \leq L_\epsilon := \max\left\{ m_o^2, \frac{(1 - m_o)^2}{m_\epsilon^2} \right\} < \infty.$$

Thus $g$ and $g'$ are uniformly bounded on $\mathcal{R}$. Finally,

$$\frac{\partial \ell(\lambda, x)}{\partial \lambda} = -g(\lambda, x), \qquad \frac{\partial g(\lambda, x)}{\partial \lambda} = g'(\lambda, x),$$

and

$$\left| \frac{\partial g'(\lambda, x)}{\partial \lambda} \right| = \frac{2|x - m_o|^3}{(1 - \lambda(x - m_o))^3} \leq \max\left\{ 2m_o^3, \frac{2(1 - m_o)^3}{m_\epsilon^3} \right\} < \infty.$$

The mean value theorem therefore gives the uniform Lipschitz property of $\ell$, $g$, and $g'$ in $\lambda$ on $\mathcal{R}$.

Next consider $\mathcal{R} = [0, \bar{\lambda}] \times [0, 1 - \kappa]$. For all $(\lambda, x) \in \mathcal{R}$,

$$1 - \lambda(x - m_o) \geq \begin{cases} 1, & x \leq m_o, \\ \dfrac{\kappa}{1 - m_o}, & x \in [m_o, 1 - \kappa]. \end{cases}$$

Hence $\ell$ is uniformly bounded on $\mathcal{R}$. Also,

$$|g(\lambda, x)| \leq \max\left\{ m_o, \frac{(1 - \kappa - m_o)(1 - m_o)}{\kappa} \right\} < \infty,$$

and

$$0 \leq g'(\lambda, x) \leq K_\kappa := \max\left\{ m_o^2, \frac{(1 - \kappa - m_o)^2}{\left( \kappa/(1 - m_o) \right)^2} \right\} < \infty.$$

Thus $g$ and $g'$ are uniformly bounded on $\mathcal{R}$. Finally,

$$\frac{\partial \ell(\lambda, x)}{\partial \lambda} = -g(\lambda, x), \qquad \frac{\partial g(\lambda, x)}{\partial \lambda} = g'(\lambda, x),$$

and

$$\left| \frac{\partial g'(\lambda, x)}{\partial \lambda} \right| = \frac{2|x - m_o|^3}{(1 - \lambda(x - m_o))^3} \leq \max\left\{ 2m_o^3, \frac{2(1 - \kappa - m_o)^3}{\left( \kappa/(1 - m_o) \right)^3} \right\} < \infty.$$

The mean value theorem gives the uniform Lipschitz property of $\ell$, $g$, and $g'$ in $\lambda$ on $\mathcal{R}$. $\square$

**Lemma A.5.** *Let $q \in \mathcal{Q}^{\mathrm{bd}}$. For $k \in \{1, 2\}$,*

$$\frac{1}{n} \sum_{i=1}^{n} \ell(\lambda_n^\star, X_i)^k \xrightarrow{\text{a.s.}} \mathbb{E}_q\big[\ell(\lambda^\star, X)^k\big] < \infty. \tag{24}$$

*In particular, taking $k = 1$ gives $\mathrm{KL}_{\inf}(\hat{q}_n, m_o) \xrightarrow{\text{a.s.}} \mathrm{KL}_{\inf}(q, m_o)$.*

*Proof.* We prove the lemma only for the case $m_o > m(q)$ (the case $m_o < m(q)$ follows analogously).

Fix $k \in \{1, 2\}$ and using triangle inequality, we have

$$\left| \frac{1}{n} \sum_{i=1}^{n} \ell(\lambda_n^\star, X_i)^k - \mathbb{E}_q[\ell(\lambda^\star, X)^k] \right| \le A_{n,k} + B_{n,k},$$

where

$$A_{n,k} := \left| \frac{1}{n} \sum_{i=1}^{n} \left( \ell(\lambda_n^\star, X_i)^k - \ell(\lambda^\star, X_i)^k \right) \right|, \quad B_{n,k} := \left| \frac{1}{n} \sum_{i=1}^{n} \ell(\lambda^\star, X_i)^k - \mathbb{E}_q[\ell(\lambda^\star, X)^k] \right|.$$

It suffices to show $A_{n,k} \to 0$ and $B_{n,k} \to 0$ almost surely.

**Step 1:** $B_{n,k} \to 0$ **a.s.** Lemma A.3 gives that

$$\mathbb{E}_q \left[ |\ell(\lambda^\star, X)|^k \right] < \infty,$$

for $k = 1, 2$. Hence SLLN holds, implying $B_{n,k} \to 0$ almost surely.

**Step 2:** $A_{n,k} \xrightarrow{\text{a.s.}} 0$ **when** $\lambda^\star \in (0, \frac{1}{1-m_o})$. Pick $\epsilon > 0$ with $I_\epsilon = [\lambda^\star - \epsilon, \lambda^\star + \epsilon] \subset (0, \bar{\lambda})$. First observe that, using Lemma A.4, $\ell$ is bounded on $I_\epsilon \times [0, 1]$ by some $C_\epsilon < \infty$. By Lemma A.6, $\lambda_n^\star \in I_\epsilon$ eventually almost surely. On that full-measure event, using Lemma A.6 and Lemma A.4, we get,

$$A_{n,1} \le \frac{1}{n} \sum_{i=1}^{n} \left| \ell(\lambda_n^\star, X_i) - \ell(\lambda^\star, X_i) \right| \le B_\epsilon |\lambda_n^\star - \lambda^\star| \xrightarrow{\text{a.s.}} 0,$$

and, using $|u^2 - v^2| \le (|u| + |v|)|u - v|$ together with boundedness of $\ell$,

$$A_{n,2} \le 2 C_\epsilon B_\epsilon |\lambda_n^\star - \lambda^\star| \xrightarrow{\text{a.s.}} 0.$$

**Step 3:** $A_{n,k} \xrightarrow{\text{a.s.}} 0$ **when** $\lambda^\star = \bar{\lambda}$. Fix $\kappa \in (0, 1)$ such that $1 - \kappa > m_o$ (since $m_o \in (0, 1)$) and split $A_{n,k} \le A_{n,k}^{(1)}(\kappa) + A_{n,k}^{(2)}(\kappa)$, where

$$A_{n,k}^{(1)}(\kappa) := \frac{1}{n} \sum_{i=1}^{n} \left| \ell(\lambda_n^\star, X_i)^k - \ell(\bar{\lambda}, X_i)^k \right| \mathbf{1}_{\{X_i \le 1-\kappa\}},$$

$$A_{n,k}^{(2)}(\kappa) := \frac{1}{n} \sum_{i=1}^{n} \left| \ell(\lambda_n^\star, X_i)^k - \ell(\bar{\lambda}, X_i)^k \right| \mathbf{1}_{\{X_i > 1-\kappa\}}.$$

*Bounding* $A_{n,k}^{(1)}(\kappa)$. For $(\lambda, x) \in [0, \bar{\lambda}] \times [0, 1-\kappa]$, using Lemma A.4, we get that $\ell$ is uniformly bounded and uniformly Lipschitz in $\lambda$ over $x \in [0, 1-\kappa]$, with constants depending only on $\kappa$ and $m_o$. We also know that $\lambda_n^\star \to \bar{\lambda}$ a.s. (see Lemma A.6). Hence, using arguments similar to Step 2, for each fixed $\kappa$ and for $k \in \{1, 2\}$, we get

$$A_{n,k}^{(1)}(\kappa) \xrightarrow{\text{a.s.}} 0.$$

*Bounding* $A_{n,k}^{(2)}(\kappa)$. For each $x \in [0, 1)$, observe that, $\frac{\partial}{\partial \lambda} \ell(\lambda, x) = -\frac{x - m_o}{1 - \lambda(x - m_o)}$. Hence $\lambda \mapsto \ell(\lambda, x)$ is increasing if $x < m_o$, decreasing if $x > m_o$, and identically zero if $x = m_o$. Since $\ell(0, x) = 0$, it follows that $\ell(\lambda, x)$ lies between 0 and $\ell(\bar{\lambda}, x)$ for all $\lambda \in [0, \bar{\lambda}]$. Therefore

$$|\ell(\lambda, x)| \le |\ell(\bar{\lambda}, x)|, \qquad \lambda \in [0, \bar{\lambda}].$$

Therefore for $k \in \{1, 2\}$,

$$\left| \ell(\lambda_n^\star, x)^k - \ell(\bar{\lambda}, x)^k \right| \le |\ell(\lambda_n^\star, x)|^k + |\ell(\bar{\lambda}, x)|^k \le 2|\ell(\bar{\lambda}, x)|^k,$$

whence

$$A_{n,k}^{(2)}(\kappa) \le \frac{2}{n} \sum_{i=1}^{n} |\ell(\bar{\lambda}, X_i)|^k \mathbf{1}_{\{X_i > 1-\kappa\}}.$$

By the SLLN, for each fixed $\kappa$,

$$\frac{1}{n}\sum_{i=1}^{n}|\ell(\bar{\lambda},X_i)|^k \mathbf{1}_{\{X_i>1-\kappa\}} \xrightarrow{\text{a.s.}} g_k(\kappa) := \mathbb{E}_q\left[|\ell(\bar{\lambda},X)|^k \mathbf{1}_{\{X>1-\kappa\}}\right].$$

*Conclusion.* Fix any sequence $\kappa_m \downarrow 0$ as $m \to \infty$. Since the countable union of null sets is null, there exists a full-measure event $\Omega_0$ on which, simultaneously for every $m \geq 1$,

$$A_{n,k}^{(1)}(\kappa_m) \to 0 \quad \text{and} \quad \frac{1}{n}\sum_{i=1}^{n}|\ell(\bar{\lambda},X_i)|^k \mathbf{1}_{\{X_i>1-\kappa_m\}} \to g_k(\kappa_m).$$

On $\Omega_0$, for every $m$,

$$\limsup_{n\to\infty} A_{n,k} \leq \limsup_{n\to\infty} A_{n,k}^{(1)}(\kappa_m) + \limsup_{n\to\infty} A_{n,k}^{(2)}(\kappa_m) \leq 2g_k(\kappa_m).$$

Since $\mathbf{1}_{\{X>1-\kappa_m\}} \downarrow 0$ pointwise and $|\ell(\bar{\lambda},X)|^k$ is integrable (Lemma A.3), dominated convergence gives $g_k(\kappa_m) \downarrow 0$. Letting $m \to \infty$ yields $\limsup_n A_{n,k} = 0$ on $\Omega_0$, i.e. $A_{n,k} \xrightarrow{\text{a.s.}} 0$.

Combining Steps 1, 2, and 3, gives (24), completing the proof.

$\square$

**Lemma A.6.** *Assume one of the following holds:*

*(i)* $\Phi(q,m_o) > 1$.

*(ii)* $\Phi(q,m_o) = 1$ *and* $\mathbb{E}_q\left[1/(1-X)^2\right] < \infty$.

*Then* $\lambda_n^\star \xrightarrow{\text{a.s.}} \lambda^\star$. *Moreover, in case (i),*

$$\sqrt{n}(\lambda_n^\star - \lambda^\star) \xrightarrow{d} \mathcal{N}\left(0, \frac{\mathrm{Var}_q\left(g(\lambda^\star,X)\right)}{\Psi'(\lambda^\star,q)^2}\right),$$

*while in case (ii),*

$$\sqrt{n}(\lambda^\star - \lambda_n^\star) \xrightarrow{d} \frac{Z_+}{\Psi'(\lambda^\star,q)},$$

*where* $Z \sim \mathcal{N}(0, \mathrm{Var}_q(g(\lambda^\star,X)))$ *and* $Z_+ := \max\{Z,0\}$.

*Proof.* First we prove the result for the case when $\Phi(q,m_o) > 1$.

**Part 1:** $\Phi(q,m_o) > 1$. We proceed in three steps.

**Step 1: Almost sure consistency of $\lambda_n^\star$.** Recall when $\Phi(q,m_o) > 1$, then there exists a unique $\lambda^\star \in (0,\bar{\lambda})$ such that, $\Psi(\lambda^\star,q) = 0$. Pick $\epsilon > 0$ with $I_\epsilon = [\lambda^\star - \epsilon, \lambda^\star + \epsilon] \subset (0,\bar{\lambda})$.

Using Lemma A.4, we get,

$$|g(\lambda,x) - g(\lambda',x)| \leq L_\epsilon |\lambda - \lambda'| \quad \text{for all } \lambda, \lambda' \in I_\epsilon.$$

Now, fix $\eta > 0$ and take a finite $\eta/(4L_\epsilon)$-net $\{\lambda_j\}_{j=1}^{J}$ of $I_\epsilon$. Since $g(\lambda_j,\cdot)$ is bounded, the SLLN gives $\Psi(\lambda_j,\hat{q}_n) \to \Psi(\lambda_j,q)$ a.s. for each $j$; thus

$$\max_{j\leq J}|\Psi(\lambda_j,\hat{q}_n) - \Psi(\lambda_j,q)| \leq \eta/2 \quad \text{for all large } n \text{ a.s.}$$

For any $\lambda \in I_\epsilon$, pick $j$ with $|\lambda - \lambda_j| \leq \eta/(4L_\epsilon)$; then

$$\left|\Psi(\lambda, \hat{q}_n) - \Psi(\lambda, q)\right| \leq \int |g(\lambda, x) - g(\lambda_j, x)| \, d\hat{q}_n(x) + \left|\Psi(\lambda_j, \hat{q}_n) - \Psi(\lambda_j, q)\right| + \int |g(\lambda, x) - g(\lambda_j, x)| \, dq(x)$$

$$\leq L_\epsilon |\lambda - \lambda_j| + \frac{\eta}{2} + L_\epsilon |\lambda - \lambda_j|$$

$$\leq 2L_\epsilon \, |\lambda - \lambda_j| + \frac{\eta}{2}$$

$$\leq 2L_\epsilon \cdot \frac{\eta}{4L_\epsilon} + \frac{\eta}{2}$$

$$= \eta.$$

Hence,

$$\sup_{\lambda \in I_\epsilon} \left|\Psi(\lambda, \hat{q}_n) - \Psi(\lambda, q)\right| \xrightarrow{\text{a.s.}} 0. \tag{ULLN-1}$$

Using Lemma A.4, we know that, $g(\lambda, x)$ and $g'(\lambda, x)$ are bounded on $I_\epsilon \times [0, 1]$, Hence, by dominated convergence theorem, we get,

$$\Psi'(\lambda, q) = E_q[g'(\lambda, X)], \quad \lambda \in I_\epsilon.$$

Using the similar arguments as above, we obtain the uniform LLN

$$\sup_{\lambda \in I_\epsilon} \left|\Psi'(\lambda, \hat{q}_n) - \Psi'(\lambda, q)\right| \xrightarrow{\text{a.s.}} 0. \tag{ULLN-2}$$

Since

$$\Psi'(\lambda, q) = \mathbb{E}_q\left[\frac{(X - m_o)^2}{(1 - \lambda(X - m_o))^2}\right] > 0 \quad \text{on } I_\epsilon,$$

the map $\Psi(\cdot, q)$ is strictly increasing on $I_\epsilon$ and we know that $\Psi(\lambda^\star, q) = 0$. Thus there exists $\upsilon_\epsilon \in (0, \epsilon)$ such that

$$\Psi(\lambda^\star - \upsilon_\epsilon, q) < 0 < \Psi(\lambda^\star + \upsilon_\epsilon, q).$$

By (ULLN-1), for a given sample path, for all large $n$,

$$\Psi(\lambda^\star - \upsilon_\epsilon, \hat{q}_n) < 0 < \Psi(\lambda^\star + \upsilon_\epsilon, \hat{q}_n).$$

By (ULLN-2), for a given sample path, for all large $n$, there exists $c_\epsilon > 0$ such that

$$\inf_{\lambda \in I_\epsilon} \Psi'(\lambda, \hat{q}_n) \geq c_\epsilon > 0,$$

so $\Psi(\cdot, \hat{q}_n)$ is eventually strictly increasing on $I_\epsilon$ on each sample path.

For the case $\Phi(q, m_o) > 1$, on each sample path, for all large $n$, we also have $\Phi(\hat{q}_n, m_o) > 1$ by the SLLN, so $\Psi(\lambda_n^\star, \hat{q}_n) = 0$ for all large $n$. The intermediate value theorem then yields, on each sample path, for all large $n$,

$$\lambda_n^\star \in (\lambda^\star - \upsilon_\epsilon, \lambda^\star + \upsilon_\epsilon) \quad \text{with} \quad \Psi(\lambda_n^\star, \hat{q}_n) = 0.$$

Letting $\epsilon \downarrow 0$ yields $\upsilon_\epsilon \downarrow 0$. Hence, it follows that,

$$\lambda_n^\star \xrightarrow{\text{a.s.}} \lambda^\star.$$

**Step 2: CLT for $\lambda_n^*$.** We now prove the second part of the Lemma. Fix $\epsilon > 0$ small. Since $\lambda \mapsto \Psi(\lambda, \hat{q}_n)$ is continuously differentiable on $I_\epsilon$, and $\lambda_n^\star \xrightarrow{\text{a.s.}} \lambda^\star$, it follows that $\lambda_n^\star \in I_\epsilon$ eventually almost surely. Applying the Mean Value Theorem to $\lambda \mapsto \Psi(\lambda, \hat{q}_n)$ along a fixed sample path, for all sufficiently large $n$ there exists

$$c_n \in \left(\min\{\lambda^\star, \lambda_n^\star\}, \max\{\lambda^\star, \lambda_n^\star\}\right)$$

such that

$$\Psi(\lambda_n^\star, \hat{q}_n) = \Psi(\lambda^\star, \hat{q}_n) + (\lambda_n^\star - \lambda^\star)\Psi'(c_n, \hat{q}_n).$$

Rearranging and multiplying by $\sqrt{n}$ gives

$$\sqrt{n}(\lambda_n^\star - \lambda^\star) = -\frac{\sqrt{n}(\Psi(\lambda^\star, \hat{q}_n) - \Psi(\lambda^\star, q))}{\Psi'(c_n, \hat{q}_n)} + \frac{\sqrt{n}\,\Psi(\lambda_n^\star, \hat{q}_n)}{\Psi'(c_n, \hat{q}_n)}, \qquad \text{since } \Psi(\lambda^\star, q) = 0. \tag{25}$$

*First term in* (25). Using Lemma A.4, for all $(\lambda, x) \in I_\epsilon \times [0, 1]$, $|g(\lambda, x)| \le B_\epsilon < \infty$. Since, we know that by definition $\lambda^\star \in I_\epsilon$, hence it following using classical CLT,

$$\sqrt{n}(\Psi(\lambda^\star, \hat{q}_n) - \Psi(\lambda^\star, q)) = \sqrt{n}\Big(\frac{1}{n}\sum_{i=1}^n g(\lambda^\star, X_i) - \mathbb{E}_q[g(\lambda^\star, X)]\Big) \overset{d}{\Rightarrow} \mathcal{N}(0, \mathrm{Var}_q(g(\lambda^\star, X))), \tag{26}$$

where $\mathrm{Var}_q(g(\lambda^\star, X)) < \infty$.

*Denominator in* (25). We show that
$$\Psi'(c_n, \hat{q}_n) \xrightarrow{\text{a.s.}} \Psi'(\lambda^\star, q) \in (0, \infty).$$

Decompose:
$$\Psi'(c_n, \hat{q}_n) - \Psi'(\lambda^\star, q) = \big(\Psi'(c_n, \hat{q}_n) - \Psi'(c_n, q)\big) + \big(\Psi'(c_n, q) - \Psi'(\lambda^\star, q)\big).$$

For the first term, the uniform LLN (ULLN-2) implies

$$\sup_{c \in I_\epsilon} |\Psi'(c, \hat{q}_n) - \Psi'(c, q)| \xrightarrow{\text{a.s.}} 0.$$

Since $c_n \to \lambda^\star$ a.s., this term converges to 0 a.s.

For the second term, note that

$$g'(\lambda, x) = \frac{(x - m_o)^2}{(1 - \lambda(x - m_o))^2}$$

is continuous in $\lambda$ and uniformly bounded by $L_\epsilon$ on $I_\epsilon \times [0, 1]$. Thus, by dominated convergence,

$$\Psi'(c_n, q) \to \Psi'(\lambda^\star, q) \quad \text{a.s.}$$

Therefore,
$$\Psi'(c_n, \hat{q}_n) \xrightarrow{\text{a.s.}} \Psi'(\lambda^\star, q). \tag{27}$$
Using the definition of $\Psi(\lambda^\star, q)$, we get that $\Psi'(\lambda^\star, q) \in (0, \infty)$.

*Second term in* (25). Recall from step 1 of this proof, on each sample path, for large $n$, $\Phi(\hat{q}_n, m_o) > 1$, hence

$$\Psi(\lambda_n^\star, \hat{q}_n) = 0.$$

Therefore, by (27),
$$\frac{\sqrt{n}\,\Psi(\lambda_n^\star, \hat{q}_n)}{\Psi'(c_n, \hat{q}_n)} \longrightarrow 0 \quad \text{a.s.} \tag{28}$$

**Step 3: Conclusion.** Combining (26), (27), and (28) and applying Slutsky's theorem yields

$$\sqrt{n}(\lambda_n^\star - \lambda^\star) \overset{d}{\Rightarrow} \mathcal{N}\Big(0, \frac{\mathrm{Var}_q(g(\lambda^\star, X))}{\Psi'(\lambda^\star, q)^2}\Big).$$

This completes the proof under the case when $\Phi(q, m_o) > 1$.

Now, we prove the results for the case when $\Phi(q, m_o) = 1$.

**Part 2:** $\Phi(q, m_o) = 1$. We proceed in five steps.

**Step 1: Almost sure consistency of $\lambda_n^\star$.** Fix any $\epsilon \in (0, \lambda^\star)$. Define

$$L(\lambda) := \mathbb{E}_q[\ell(\lambda, X)].$$

Observe that $L$ is continuous and increasing on $[0, \bar{\lambda}]$ and has a unique maximizer at $\lambda^\star$ (hence $L(\lambda^\star) > L(\lambda^\star - \epsilon)$), there exists $\eta > 0$ such that

$$L(\lambda^\star) - L(\lambda^\star - \epsilon) \geq 4\eta > 0. \tag{29}$$

Moreover, on the compact interval $[0, \lambda^\star - \epsilon]$, the function $\ell(\lambda, x)$ is bounded and continuously differentiable in $\lambda$ for every $x \in [0, 1]$. Hence, by the same arguments used to establish ULLN-1, we get,

$$\sup_{\lambda \in [0, \lambda^\star - \epsilon]} \left| \mathbb{E}_{\hat{q}_n}[\ell(\lambda, X)] - L(\lambda) \right| \xrightarrow{\text{a.s.}} 0.$$

In addition, using SLLN we have,

$$\mathbb{E}_{\hat{q}_n}[\ell(\lambda^\star, X)] \xrightarrow{\text{a.s.}} L(\lambda^\star).$$

Hence, for all sufficiently large $n$ (a.s.),

$$\mathbb{E}_{\hat{q}_n}[\ell(\lambda^\star, X)] \geq L(\lambda^\star) - \eta,$$
$$\sup_{\lambda \in [0, \lambda^\star - \epsilon]} \mathbb{E}_{\hat{q}_n}[\ell(\lambda, X)] \leq \sup_{\lambda \in [0, \lambda^\star - \epsilon]} L(\lambda) + \eta = L(\lambda^\star - \epsilon) + \eta,$$

where the last equality uses that $L$ is increasing on $[0, \lambda^\star]$. Combining the last two displays with (29) yields, for all sufficiently large $n$ (a.s.),

$$\mathbb{E}_{\hat{q}_n}[\ell(\lambda^\star, X)] \geq \sup_{\lambda \in [0, \lambda^\star - \epsilon]} \mathbb{E}_{\hat{q}_n}[\ell(\lambda, X)] \geq \mathbb{E}_{\hat{q}_n}[\ell(\lambda, X)] \qquad \forall \lambda \in [0, \lambda^\star - \epsilon].$$

Hence, it follows that any maximizer $\lambda_n^\star$ must satisfy $\lambda_n^\star > \lambda^\star - \epsilon$. As $\epsilon \downarrow 0$ and always $\lambda_n^\star \leq \lambda^\star$, this gives $\lambda_n^\star \xrightarrow{\text{a.s.}} \lambda^\star$.

**Step 2: Taylor series expansion.** Consider the following function:

$$N(X_1, X_2, \ldots X_n, \lambda_n^\star) = \sum_{i=1}^n \big( g(\lambda_n^\star, X_i) - g(\lambda^\star, X_i) \big).$$

Since $\Phi(q, m_o) = 1$, it follows that $q\{1\} = 0$, and hence we may restrict the analysis to the measure-one event $\{X_i \in [0, 1) \; \forall i = 1, 2, 3, \ldots\}$. Using the definition of $g(\lambda, X)$, $N(X_1, X_2, \ldots X_n, \lambda_n^\star)$ is continuously differentiable in $\lambda_n^\star$ for any values of $X_1, X_2, \ldots X_n \in [0, 1)$ and any value of $\lambda_n^\star \in \left[0, \frac{1}{1 - m_o}\right]$. Hence, using Taylor expansion of $N(X_1, X_2, \ldots X_n, \lambda_n^\star)$ in $\lambda_n^\star$ around $\lambda^\star$ yields

$$N(X_1, X_2, \ldots X_n, \lambda_n^\star) = (\lambda_n^\star - \lambda^\star) \frac{\partial N(X_1, X_2, \ldots X_n, \lambda)}{\partial \lambda} \bigg|_{\lambda = c(X_1, X_2, \ldots X_n, \lambda_n^\star)}.$$

For ease of notation, let $c_n = c(X_1, X_2, \ldots X_n, \lambda_n^\star)$. Observe that

$$\Psi(\lambda_n^\star, \hat{q}_n) - \Psi(\lambda^\star, \hat{q}_n) = \frac{N(X_1, X_2, \ldots X_n, \lambda_n^\star)}{n}.$$

Hence, it follows that following holds:

$$\Psi(\lambda_n^\star, \hat{q}_n) - \Psi(\lambda^\star, \hat{q}_n) = \Psi'(c_n, \hat{q}_n) (\lambda_n^\star - \lambda^\star),$$

for some $c_n$ between $\lambda_n^\star$ and $\lambda^\star$.

Observe that, for a given $X_1, X_2, \ldots X_n \in [0, 1)$, $\mathbb{E}_{\hat{q}_n}[\ell(\lambda, X)]$ is continuously differentiable in $\lambda$ for $\lambda \in \left[0, \frac{1}{1 - m_o}\right]$. Since $\lambda_n^\star$ maximizes $\mathbb{E}_{\hat{q}_n}[\ell(\lambda, X)]$ on $(0, \lambda^\star]$ and this objective is concave in $\lambda$, we have the usual first-order optimality conditions:

- If $\lambda_n^\star \in (0, \lambda^\star)$, then

$$\frac{\partial}{\partial \lambda} \mathbb{E}_{\hat{q}_n}[\ell(\lambda, X)]\Big|_{\lambda = \lambda_n^\star} = -\Psi(\lambda_n^\star, \hat{q}_n) = 0,$$

and monotonicity in $\lambda$ gives $\Psi(\lambda^\star, \hat{q}_n) \geq 0$.

- If $\lambda_n^\star = \lambda^\star$, we have,

$$\frac{\partial}{\partial \lambda} \mathbb{E}_{\hat{q}_n}[\ell(\lambda, X)]\Big|_{\lambda = \lambda^\star} \geq 0, \quad \text{i.e. } -\Psi(\lambda^\star, \hat{q}_n) \geq 0 \implies \Psi(\lambda^\star, \hat{q}_n) \leq 0.$$

Thus, when $\lambda_n^\star \in (0, \lambda^\star)$,

$$\lambda^\star - \lambda_n^\star = \frac{\Psi(\lambda^\star, \hat{q}_n)}{\Psi'(c_n, \hat{q}_n)}, \qquad \Psi(\lambda^\star, \hat{q}_n) \geq 0,$$

whereas when $\lambda_n^\star = \lambda^\star$, we trivially have

$$\lambda^\star - \lambda_n^\star = 0 \qquad \text{and} \qquad \Psi(\lambda^\star, \hat{q}_n) \leq 0.$$

Hence, following holds:

$$\lambda^\star - \lambda_n^\star = \frac{\left[\Psi(\lambda^\star, \hat{q}_n)\right]_+}{\Psi'(c_n, \hat{q}_n)}, \qquad [x]_+ := \max\{x, 0\}. \tag{30}$$

**Step 3: Convergence of the denominator in** (30)**.** Fix $\kappa \in (0, 1)$ and decompose

$$\Psi'(c_n, \hat{q}_n) - \mathbb{E}_q[g'(\lambda^\star, X)] = \Big(\mathbb{E}_{\hat{q}_n}[g'(c_n, X)] - \mathbb{E}_q[g'(c_n, X)]\Big) + \Big(\mathbb{E}_q[g'(c_n, X)] - \mathbb{E}_q[g'(\lambda^\star, X)]\Big)$$

$$=: A_n + B_n.$$

**Step 3.1:** $B_n \to 0$ **a.s.** Since $c_n \to \lambda^\star$ a.s. and for each $x < 1$ the map $\lambda \mapsto g'(\lambda, x)$ is continuous, we have $g'(c_n, x) \to g'(\lambda^\star, x)$ pointwise for all $x < 1$. Moreover, for any $\lambda \in [0, \frac{1}{1-m_o}]$ and $x \in [0, 1)$,

$$0 \leq g'(\lambda, x) = \frac{(x - m_o)^2}{(1 - \lambda(x - m_o))^2} \leq (1 - m_o)^2 \frac{1}{(1-x)^2}.$$

The function $(1 - m_o)^2 (1 - x)^{-2}$ is integrable under $\mathbb{E}_q[(1 - X)^{-2}] < \infty$. Hence, by dominated convergence,

$$B_n = \mathbb{E}_q[g'(c_n, X) - g'(\lambda^\star, X)] \xrightarrow{\text{a.s.}} 0.$$

**Step 3.2:** $A_n \to 0$ **a.s.** Write $A_n = A_{n,\kappa}^{(1)} + A_{n,\kappa}^{(2)}$, where

$$A_{n,\kappa}^{(1)} := \mathbb{E}_{\hat{q}_n}[g'(c_n, X)\mathbf{1}_{\{X \leq 1-\kappa\}}] - \mathbb{E}_q[g'(c_n, X)\mathbf{1}_{\{X \leq 1-\kappa\}}],$$

$$A_{n,\kappa}^{(2)} := \mathbb{E}_{\hat{q}_n}[g'(c_n, X)\mathbf{1}_{\{X > 1-\kappa\}}] - \mathbb{E}_q[g'(c_n, X)\mathbf{1}_{\{X > 1-\kappa\}}].$$

*(i) Handling $A_{n,\kappa}^{(1)}$.* Using Lemma A.4, we have, $\{g'(\lambda, \cdot)\mathbf{1}_{\{\cdot \leq 1-\kappa\}} : \lambda \in [0, \lambda^\star]\}$ is uniformly bounded and Lipschitz in $\lambda$. Hence, using arguments similar to show ULLN-1, we get,

$$\sup_{\lambda \in [0, \lambda^\star]} \left|\mathbb{E}_{\hat{q}_n}[g'(\lambda, X)\mathbf{1}_{\{X \leq 1-\kappa\}}] - \mathbb{E}_q[g'(\lambda, X)\mathbf{1}_{\{X \leq 1-\kappa\}}]\right| \xrightarrow{\text{a.s.}} 0,$$

hence $A_{n,\kappa}^{(1)} \to 0$ a.s.

*(ii) Handling $A_{n,\kappa}^{(2)}$.* For all $\lambda \in [0, \lambda^\star]$ and $x \in [0, 1)$,

$$0 \leq g'(\lambda, x)\mathbf{1}_{\{x > 1-\kappa\}} \leq (1 - m_o)^2 \frac{1}{(1-x)^2}\mathbf{1}_{\{x > 1-\kappa\}} =: (1 - m_o)^2 H_\kappa(x).$$

By the SLLN,

$$\mathbb{E}_{\hat{q}_n}[H_\kappa(X)] \xrightarrow{\text{a.s.}} \mathbb{E}_q[H_\kappa(X)].$$

Therefore,

$$\limsup_{n\to\infty} |A^{(2)}_{n,\kappa}| \leq 2(1 - m_o)^2 \, \mathbb{E}_q[H_\kappa(X)] \quad \text{a.s.}$$

Since $H_\kappa(X) \downarrow 0$ a.s. as $\kappa \downarrow 0$ and $0 \leq H_\kappa(X) \leq (1 - X)^{-2}$ with $\mathbb{E}_q[(1 - X)^{-2}] < \infty$, dominated convergence yields $\mathbb{E}_q[H_\kappa(X)] \to 0$. Letting $\kappa \downarrow 0$ gives $A_n \to 0$ a.s.

Combining Steps 3.1 and 3.2 yields

$$\Psi'(c_n, \hat{q}_n) = \mathbb{E}_{\hat{q}_n}[g'(c_n, X)] \xrightarrow{\text{a.s.}} \mathbb{E}_q[g'(\lambda^\star, X)].$$

Applying the same arguments as in Step 3.1 of this proof, via the dominated convergence theorem, we get $\mathbb{E}_q[g'(\lambda^\star, X)] = \Psi'(\lambda^\star, q)$.

**Step 4: CLT for the numerator in** (30)**.** Define

$$\alpha_n := \Psi(\lambda^\star, \hat{q}_n) - \Psi(\lambda^\star, q).$$

Note that $\Psi(\lambda^\star, q) = 0$, since $\Phi(q, m_o) = 1$. Using,

$$g(\lambda^\star, x) = (1 - m_o)\frac{x - m_o}{1 - x}, \qquad g(\lambda^\star, x)^2 \leq (1 - m_o)^2(1 - x)^{-2},$$

the assumption $\mathbb{E}_q[(1 - X)^{-2}] < \infty$ implies $\mathbb{E}_q[g(\lambda^\star, X)^2] < \infty$. Thus, by the classical CLT,

$$\sqrt{n}\,\alpha_n = \sqrt{n}\Big(\frac{1}{n}\sum_{i=1}^n g(\lambda^\star, X_i) - \mathbb{E}_q[g(\lambda^\star, X)]\Big) \xRightarrow{d} \mathcal{N}\big(0, \, \mathrm{Var}_q\big(g(\lambda^\star, X)\big)\big),$$

with $\mathrm{Var}_q\big(g(\lambda^\star, X)\big) < \infty$.

**Step 5: Conclusion.** From (30),

$$\sqrt{n}(\lambda^\star - \lambda^\star_n) = \frac{\big[\sqrt{n}\Psi(\lambda^\star, \hat{q}_n)\big]_+}{\Psi'(c_n, \hat{q}_n)} = \frac{\big[\sqrt{n}\alpha_n\big]_+}{\Psi'(c_n, \hat{q}_n)}.$$

By Step 3, $\Psi'(c_n, \hat{q}_n) \to \Psi'(\lambda^\star, q) \in (0, \infty)$ a.s., and by Step 4, $\sqrt{n}\alpha_n \xRightarrow{d} \mathcal{N}\big(0, \mathrm{Var}_q\big(g(\lambda^\star, X)\big)\big)$. By Slutsky's theorem and continuity of $z \mapsto z_+$,

$$\sqrt{n}(\lambda^\star - \lambda^\star_n) \xRightarrow{d} \frac{Z_+}{\Psi'(\lambda^\star, q)},$$

where $Z \sim \mathcal{N}\big(0, \mathrm{Var}_q\big(g(\lambda^\star, X)\big)\big)$. This completes the proof. $\qquad\square$ $\hfill\square$

**Lemma A.7.** *When $\Phi(q, m_o) \geq 1$, for $c_n \in [0, 1/(1 - m_o)]$ with $c_n \to \lambda^\star$ almost surely, the following holds:*

$$A_n := \frac{1}{n}\sum_{i=1}^n g(c_n, X_i) \xrightarrow{\text{a.s.}} 0.$$

*Proof.* Observe that,

$$A_n = \underbrace{\frac{1}{n}\sum_{i=1}^n g(\lambda^\star, X_i)}_{\text{Term 1}} + \underbrace{\frac{1}{n}\sum_{i=1}^n \Big(g(c_n, X_i) - g(\lambda^\star, X_i)\Big)}_{\text{Term 2}}.$$

Below, we show that each of the two terms above converge almost surely to 0, giving the desired convergence result.

**Analyzing Term 1.** By the SLLN, we have

$$\frac{1}{n}\sum_{i=1}^{n} g(\lambda^\star, X_i) \xrightarrow{\text{a.s.}} \Psi(\lambda^\star, q) = 0, \tag{31}$$

where the last equality follows from the definition of $\lambda^*$ when $\Phi(q, m_o) \geq 1$ holds.

**Analyzing Term 2.** We treat the two cases, $\Phi(q, m_o) > 1$ and $\Phi(q, m_o) = 1$, separately. Consider the probability 1 set

$$\Omega_0 := \left\{ \lim_{n\to\infty} c_n = \lambda^\star \right\}.$$

*Case (A):* $\Phi(q, m_o) > 1$. In this case, we have, $\lambda^\star < \bar\lambda$. Fix any $\epsilon > 0$ with $I_\epsilon = [\lambda^\star - \epsilon, \ \lambda^\star + \epsilon] \subset (0, \bar\lambda)$. Next, fix $\omega \in \Omega_0$. There exists $N(\omega)$ such that $c_n(\omega) \in I_\epsilon$ for all $n \geq N(\omega)$. For such $n$, using Lemma A.4,

$$\left| g(c_n, X_i) - g(\lambda^\star, X_i) \right| \leq L_\epsilon |c_n - \lambda^\star| \quad \text{for each } i.$$

Hence,

$$\left| \frac{1}{n}\sum_{i=1}^{n} \left( g(c_n, X_i) - g(\lambda^\star, X_i) \right) \right| \leq L_\epsilon |c_n - \lambda^\star| \xrightarrow[n\to\infty]{} 0 \quad \text{a.s.}$$

*Case (B):* $\Phi(q, m_o) = 1$. Recall that in this case, $\lambda^\star = 1/(1 - m_o)$ and $q(\{1\}) = 0$. Fix $\kappa \in (0,1)$ such that $1 - \kappa > m_o$ (since $m_o \in (0,1)$), and split

$$\frac{1}{n}\sum_{i=1}^{n} \left( g(c_n, X_i) - g(\lambda^\star, X_i) \right)$$

$$= \underbrace{\frac{1}{n}\sum_{i=1}^{n} \mathbf{1}_{\{X_i \leq 1-\kappa\}} \left( g(c_n, X_i) - g(\lambda^\star, X_i) \right)}_{\text{Term A}} + \underbrace{\frac{1}{n}\sum_{i=1}^{n} \mathbf{1}_{\{X_i > 1-\kappa\}} \left( g(c_n, X_i) - g(\lambda^\star, X_i) \right)}_{\text{Term B}}.$$

Using Lemma A.4, we get,

$$|g(\lambda, x) - g(\lambda', x)| \leq K_\kappa |\lambda - \lambda'| \quad \text{for all } \lambda, \lambda' \in \left[0, \frac{1}{1-m_o}\right], \text{ and } x \in [0, 1-\kappa].$$

Hence

$$\left| \frac{1}{n}\sum_{i=1}^{n} \mathbf{1}_{\{X_i \leq 1-\kappa\}} \left( g(c_n, X_i) - g(\lambda^\star, X_i) \right) \right| \leq K_\kappa |c_n - \lambda^\star| \xrightarrow[n\to\infty]{\text{a.s.}} 0. \tag{32}$$

Next, since $q(\{1\}) = 0$, we have $\{1 > X > 1 - \kappa\} = \{1 \geq X > 1 - \kappa\}$ almost surely, where $X \sim q$. Now, for any $c \in [0, 1/(1-m_o)]$ and $m_o < 1 - \kappa < x < 1$,

$$|g(c, x)| = \frac{|x - m_o|}{1 - c(x - m_o)} \leq \frac{|x - m_o|}{1 - \frac{1}{1-m_o}(x - m_o)} = \frac{(1 - m_o)|x - m_o|}{1 - x}.$$

Hence, we have

$$\left| \frac{1}{n}\sum_{i=1}^{n} \mathbf{1}_{\{X_i > 1-\kappa\}} \left( g(c_n, X_i) - g(\lambda^\star, X_i) \right) \right| \leq \frac{1}{n}\sum_{i=1}^{n} \mathbf{1}_{\{1 > X_i > 1-\kappa\}} \left( |g(c_n, X_i)| + |g(\lambda^\star, X_i)| \right)$$

$$\leq \frac{2(1 - m_o)}{n}\sum_{i=1}^{n} \mathbf{1}_{\{1 > X_i > 1-\kappa\}} \frac{|X_i - m_o|}{1 - X_i}. \tag{33}$$

Define

$$Y_\kappa(X) := \frac{|X - m_o|}{1 - X} \mathbf{1}_{\{1 > X > 1 - \kappa\}}.$$

From the bound above,

$$\left| \frac{1}{n} \sum_{i=1}^{n} \mathbf{1}_{\{X_i > 1 - \kappa\}} \Big( g(c_n, X_i) - g(\lambda^\star, X_i) \Big) \right| \leq \frac{2(1 - m_o)}{n} \sum_{i=1}^{n} Y_\kappa(X_i).$$

Observe that,

$$0 \leq Y_\kappa(X) \leq \frac{(1 - m_o)}{(1 - X)}.$$

Further, $(1 - m_o)/(1 - X)$ is integrable since we have assumed that $\mathbb{E}_q[1/(1-X)^2] < \infty$. Hence, by SLLN,

$$\frac{1}{n} \sum_{i=1}^{n} Y_\kappa(X_i) \xrightarrow[n \to \infty]{a.s.} \mathbb{E}_q[Y_\kappa(X)].$$

Moreover, $Y_\kappa(X) \downarrow 0$ a.s. as $\kappa \downarrow 0$ hence by dominated convergence,

$$\mathbb{E}_q[Y_\kappa(X)] \to 0.$$

Therefore,

$$\lim_{\kappa \downarrow 0} \lim_{n \to \infty} \left| \frac{1}{n} \sum_{i=1}^{n} \mathbf{1}_{\{X_i > 1 - \kappa\}} \Big( g(c_n, X_i) - g(\lambda^\star, X_i) \Big) \right| = 0 \quad \text{a.s.} \tag{34}$$

Combining the bounds from (32) and (34), we get

$$\frac{1}{n} \sum_{i=1}^{n} \Big( g(c_n, X_i) - g(\lambda^\star, X_i) \Big) \to 0 \quad \text{a.s.} \tag{35}$$

Combining (31) and (35), we get the desired result, completing the proof.

$\square$

**Lemma A.8.** *Let* $L_n := \mathrm{KL}_{\inf}(\hat{q}_n, m_o)$ *and* $L := \mathrm{KL}_{\inf}(q, m_o)$. *Define*

$$Y_n := \sqrt{n}(L_n - L) = \sqrt{n}(\mathrm{KL}_{\inf}(\hat{q}_n, m_o) - \mathrm{KL}_{\inf}(q, m_o)).$$

*Then* $\{Y_n\}$ *satisfies the Anscombe condition, that is, for every* $\epsilon > 0$ *and* $\eta > 0$, *there exist* $\delta \in (0, 1)$ *and* $n_0 \geq 1$ *such that for all* $n \geq n_0$,

$$\mathbb{P} \left( \max_{\substack{k \in \mathbb{N}: \\ |k-n| \leq n\delta}} |Y_k - Y_n| > \epsilon \right) < \eta.$$

*Proof.* As earlier, we only prove the result for the case $m_o > m(q)$. The other case ($m_o < m(q)$) follows analogously. Fix $\epsilon > 0$ and $\eta > 0$. The proof proceeds in 7 steps, detailed below.

**Step 1.** For any $\delta \in (0, 1)$ and integer $k$ with $|k - n| \leq n\delta$,

$$Y_k - Y_n = \sqrt{n} \left( (L_k - L_n) + \left( \sqrt{k/n} - 1 \right) (L_k - L) \right).$$

Therefore,

$$\max_{|k-n| \leq n\delta} |Y_k - Y_n| \leq \sqrt{n} \max_{|k-n| \leq n\delta} |L_k - L_n| + \sqrt{n} \left( \max_{|k-n| \leq n\delta} |\sqrt{k/n} - 1| \right) \left( \max_{|k-n| \leq n\delta} |L_k - L| \right)$$

$$\leq \sqrt{n} \max_{|k-n| \leq n\delta} |L_k - L_n| + \sqrt{n}\delta \left( \max_{|k-n| \leq n\delta} |L_k - L| \right), \tag{36}$$

where the last inequality follows since $|k - n| \leq n\delta$ implies $k/n \in [1 - \delta, 1 + \delta]$, and we have

$$\max_{|k-n|\leq n\delta} \left|\sqrt{k/n} - 1\right| = \max\{1 - \sqrt{1 - \delta}, \sqrt{1 + \delta} - 1\} = 1 - \sqrt{1 - \delta} \leq \delta.$$

Define

$$M_{n,\delta} := \max_{|k-n|\leq n\delta} \sqrt{n} \, |L_k - L|.$$

Using (36) along with a union bound, we get that for any $\delta \in (0, 1)$, the following inequality holds:

$$\mathbb{P}\left(\max_{|k-n|\leq n\delta} |Y_k - Y_n| > \epsilon\right) \leq \mathbb{P}\left(\sqrt{n} \max_{|k-n|\leq n\delta} |L_k - L_n| > \epsilon/2\right) + \mathbb{P}\left(M_{n,\delta} > \frac{\epsilon}{2\delta}\right). \tag{37}$$

Thus it suffices to prove that there exists $\delta \in (0, 1)$ and $n_0$ such that for all $n \geq n_0$ the following hold:

$$\mathbb{P}\left(\sqrt{n} \max_{|k-n|\leq n\delta} |L_k - L_n| > \frac{\epsilon}{2}\right) < \eta/2, \tag{A}$$

and

$$\mathbb{P}\left(M_{n,\delta} > \frac{\epsilon}{2\delta}\right) \leq \eta/2. \tag{B}$$

We prove existence of $\delta$ and $n_0$ such that (A) and (B) hold, below.

**Step 2: three-term decomposition of $L_k - L_n$.** Add and subtract $\ell(\lambda^\star, \cdot)$.

$$L_k - L_n = \underbrace{\frac{1}{k}\sum_{i=1}^{k}\left(\ell(\lambda_k^\star, X_i) - \ell(\lambda^\star, X_i)\right)}_{=:U_k} \underbrace{- \frac{1}{n}\sum_{i=1}^{n}\left(\ell(\lambda_n^\star, X_i) - \ell(\lambda^\star, X_i)\right)}_{=:U_n}$$

$$+ \underbrace{\left(\frac{1}{k}\sum_{i=1}^{k}\ell(\lambda^\star, X_i) - \frac{1}{n}\sum_{i=1}^{n}\ell(\lambda^\star, X_i)\right)}_{=:V_{k,n}}. \tag{38}$$

Hence,

$$\sqrt{n}\max_{|k-n|\leq n\delta}|L_k - L_n| \leq \underbrace{\sqrt{n}\max_{|k-n|\leq n\delta}|U_k|}_{\text{Term 1}} + \underbrace{\sqrt{n}|U_n|}_{\text{Term 2}} + \underbrace{\sqrt{n}\max_{|k-n|\leq n\delta}|V_{k,n}|}_{\text{Term 3}}. \tag{39}$$

**Step 3: control Term 3.**

Let

$$Z_i := \ell(\lambda^\star, X_i) - \mathbb{E}[\ell(\lambda^\star, X)], \qquad S_t := \sum_{i=1}^{t} Z_i.$$

Then $\mathbb{E}[Z_i] = 0$. In addition, $\mathrm{Var}(Z_1) < \infty$ (Lemma A.3).

Since

$$\frac{1}{t}\sum_{i=1}^{t}\ell(\lambda^\star, X_i) - \mathbb{E}[\ell(\lambda^\star, X)] = \frac{S_t}{t},$$

we have

$$V_{k,n} = \frac{S_k}{k} - \frac{S_n}{n} = \frac{S_k - S_n}{k} + S_n\left(\frac{1}{k} - \frac{1}{n}\right).$$

For $|k - n| \leq n\delta$, we have $k \geq (1 - \delta)n$ and $|n - k| \leq n\delta$, hence

$$\sqrt{n}|V_{k,n}| \leq \frac{\sqrt{n}}{k}|S_k - S_n| + \sqrt{n}|S_n|\frac{|n - k|}{kn} \leq \frac{1}{(1-\delta)\sqrt{n}}|S_k - S_n| + \frac{\delta}{(1-\delta)\sqrt{n}}|S_n|.$$

Therefore, with $a := \epsilon/6$,

$$\mathbb{P}\left(\sqrt{n} \max_{|k-n|\leq n\delta} |V_{k,n}| > a\right) \leq \mathbb{P}\left(\max_{|k-n|\leq n\delta} |S_k - S_n| > \frac{a}{2}(1-\delta)\sqrt{n}\right)$$
$$+ \mathbb{P}\left(|S_n| > \frac{a}{2}\frac{1-\delta}{\delta}\sqrt{n}\right). \tag{40}$$

Let $\sigma_Z^2 := \mathrm{Var}(Z_1)$.

For the increment term, set $N := \lfloor n\delta \rfloor$ and note that

$$\max_{|k-n|\leq N} |S_k - S_n| = \max\left\{\max_{1\leq j\leq N}\left|\sum_{i=n+1}^{n+j} Z_i\right|, \max_{1\leq j\leq N}\left|\sum_{i=n-j+1}^{n} Z_i\right|\right\}.$$

Hence, for any $u > 0$,

$$\mathbb{P}\left(\max_{|k-n|\leq N} |S_k - S_n| > u\right) \leq \mathbb{P}\left(\max_{1\leq j\leq N}\left|\sum_{i=n+1}^{n+j} Z_i\right| > u\right) + \mathbb{P}\left(\max_{1\leq j\leq N}\left|\sum_{i=n-j+1}^{n} Z_i\right| > u\right).$$

By stationarity, each term has the same bound as $\mathbb{P}\left(\max_{1\leq j\leq N}\left|\sum_{i=1}^{j} Z_i\right| > u\right)$. By Kolmogorov's maximal inequality (Billingsley, 2017, Theorem 22.4), for any $u > 0$,

$$\mathbb{P}\left(\max_{1\leq j\leq N}\left|\sum_{i=1}^{j} Z_i\right| \geq u\right) \leq \frac{N\sigma_Z^2}{u^2}.$$

Taking $u = \frac{a}{2}(1-\delta)\sqrt{n}$ yields

$$\mathbb{P}\left(\max_{|k-n|\leq n\delta} |S_k - S_n| > \frac{a}{2}(1-\delta)\sqrt{n}\right) \leq 2 \cdot \frac{(n\delta)\sigma_Z^2}{(\frac{a}{2}(1-\delta)\sqrt{n})^2} = \frac{8\sigma_Z^2}{a^2(1-\delta)^2}\delta. \tag{41}$$

For the second term in (40), using Chebyshev inequality,

$$\mathbb{P}\left(|S_n| > \frac{a}{2}\frac{1-\delta}{\delta}\sqrt{n}\right) \leq \frac{n\sigma_Z^2}{(\frac{a}{2}\frac{1-\delta}{\delta}\sqrt{n})^2} = \frac{4\sigma_Z^2}{a^2(1-\delta)^2}\delta^2. \tag{42}$$

Combining (40)–(41)–(42) yields the explicit bound

$$\mathbb{P}\left(\sqrt{n} \max_{|k-n|\leq n\delta} |V_{k,n}| > \epsilon/6\right) \leq C_V\,\delta + C_V'\,\delta^2, \tag{43}$$

for constants $C_V, C_V'$ depending only on $\sigma_Z^2$ and $\epsilon$ (and not on $n$).

**Step 4: case analysis for Terms 1 and 2 .**

**Case 1:** $\Phi(q, m_o) < 1$. In this case, $\lambda^\star = \bar{\lambda}$. By the SLLN,

$$\Phi(\hat{q}_n, m_o) = \frac{1}{n}\sum_{i=1}^{n} \frac{1-m_o}{1-X_i} \xrightarrow{\text{a.s.}} \Phi(q, m_o) < 1.$$

Hence, on each sample path there exists $N_0(\omega)$ such that for all $t \geq N_0(\omega)$, $\Phi(\hat{q}_t, m_o) < 1$, which implies $\lambda_t^\star = \bar{\lambda} = \lambda^\star$ for all $t \geq N_0(\omega)$. Therefore, it follows that $\sqrt{n}\max_{|k-n|\leq n\delta}|U_k| \xrightarrow{p} 0$ and $\sqrt{n}|U_n| \xrightarrow{p} 0$.

**Case 2 and 3:** $\Phi(q, m_o) \geq 1$.

Observe that this $U_n$ is same as $T_{1,n}$ defined in the proof of Theorem 4.2. Hence using Taylor series it follows that, we have

$$\sqrt{n}|U_n| = \sqrt{n}\,|\lambda_n^\star - \lambda^\star|\,|A_n|, \qquad \sqrt{n}\max_{|k-n|\leq n\delta}|U_k| \leq \frac{1}{\sqrt{1-\delta}}\Big(\max_{|k-n|\leq n\delta}\sqrt{k}\,|\lambda_k^\star - \lambda^\star|\Big)\Big(\max_{|k-n|\leq n\delta}|A_k|\Big),$$

because $k \geq (1-\delta)n$ implies $\sqrt{n} \leq \sqrt{k}/\sqrt{1-\delta}$. Here,

$$A_n := \frac{1}{n}\sum_{i=1}^{n} g(c_n, X_i),$$

for some $c_n$ between $\lambda_n^\star$ and $\lambda^\star$. Thus, to control Terms 1–2 we will show (in each case):

$$\max_{|k-n|\leq n\delta}|A_k| \to 0 \text{ (a.s.)} \quad \text{and} \quad \max_{|k-n|\leq n\delta}\sqrt{k}\,|\lambda_k^\star - \lambda^\star| = O_p(1). \tag{44}$$

**Case 2:** $\Phi(q, m_o) > 1$. In this case, $\lambda^\star < \bar{\lambda}$ and it is the unique solution of $\Psi(\lambda^\star, q) = 0$.

For Term 2, we have

$$\sqrt{n}|U_n| = \sqrt{n}\,|\lambda_n^\star - \lambda^\star|\,|A_n|.$$

By Lemma A.6 amd Lemma A.7, $\sqrt{n}(\lambda_n^\star - \lambda^\star) = O_p(1)$ and, $A_n \to 0$ a.s., hence $\sqrt{n}|U_n| \to 0$ in probability.

For Term 1, it is enough to show: (i) $\max_{|k-n|\leq n\delta}|A_k| \to 0$ a.s., and (ii) $\max_{|k-n|\leq n\delta}\sqrt{k}|\lambda_k^\star - \lambda^\star| = O_p(1)$. Property (i) follows from $A_k \to 0$ a.s. and the fact that $\sup_{k\geq(1-\delta)n}|A_k| \to 0$ a.s.

For (ii), using (25), we have

$$\sqrt{k}(\lambda_k^\star - \lambda^\star) = -\frac{\sqrt{k}\big(\Psi(\lambda^\star, \hat{q}_k) - \Psi(\lambda^\star, q)\big)}{\Psi'(c_k, \hat{q}_k)} + \frac{\sqrt{k}\,\Psi(\lambda_k^\star, \hat{q}_k)}{\Psi'(c_k, \hat{q}_k)}, \qquad \text{since } \Psi(\lambda^\star, q) = 0, \tag{45}$$

for some $c_k$ between $\lambda_k^\star$ and $\lambda^\star$.

Recall that under this case, for all sufficiently large $k$, $\Psi(\lambda_k^\star, \hat{q}_k) = 0$, and that $\Psi'(c_k, \hat{q}_k) \to \Psi'(\lambda^\star, q) \in (0, \infty)$ almost surely (see Proof of Lemma A.6). First, this implies that $\Psi'(c_k, \hat{q}_k)$ is bounded below by a positive constant $1/C$ for all sufficiently large $k$ on each sample path.

Consequently, we obtain

$$\sqrt{k}\,|\lambda_k^\star - \lambda^\star| \leq C\left|\frac{1}{\sqrt{k}}\sum_{i=1}^{k}\big(g(\lambda^\star, X_i) - \mathbb{E}[g(\lambda^\star, X)]\big)\right|.$$

Let $\widetilde{Z}_i := g(\lambda^\star, X_i) - \mathbb{E}[g(\lambda^\star, X)]$ and $\widetilde{S}_t := \sum_{i=1}^{t}\widetilde{Z}_i$. Then $\mathbb{E}[\widetilde{Z}_i] = 0$ and $\text{Var}(\widetilde{Z}_1) < \infty$ (see Proof of Lemma A.6). By Kolmogorov's maximal inequality, for any $M > 0$ and any $n \geq 1$,

$$\mathbb{P}\left(\max_{t\leq 2n}|\widetilde{S}_t| > M\sqrt{n}\right) \leq \frac{\text{Var}(\widetilde{S}_{2n})}{M^2 n} = \frac{2n\,\text{Var}(\widetilde{Z}_1)}{M^2 n} = \frac{2\,\text{Var}(\widetilde{Z}_1)}{M^2}.$$

For all $k$ with $|k-n| \leq n\delta$ we have $k \geq (1-\delta)n$ and $k \leq 2n$, hence

$$\max_{|k-n|\leq n\delta}\frac{|\widetilde{S}_k|}{\sqrt{k}} \leq \frac{1}{\sqrt{(1-\delta)n}}\max_{t\leq 2n}|\widetilde{S}_t|.$$

Therefore, for any $M > 0$ and any $n \geq 1$,

$$\mathbb{P}\left(\max_{|k-n|\leq n\delta}\frac{|\widetilde{S}_k|}{\sqrt{k}} > M\right) \leq \mathbb{P}\left(\max_{t\leq 2n}|\widetilde{S}_t| > M\sqrt{(1-\delta)n}\right) \leq \frac{2\,\text{Var}(\widetilde{Z}_1)}{(1-\delta)M^2}.$$

Combining this with the display above yields, for any $M > 0$,

$$\mathbb{P}\left(\max_{|k-n|\leq n\delta} \sqrt{k}\,|\lambda_k^\star - \lambda^\star| > CM\right) \leq \mathbb{P}\left(\max_{|k-n|\leq n\delta} \frac{|\widetilde{S}_k|}{\sqrt{k}} > M\right) \leq \frac{2\,\mathrm{Var}(\widetilde{Z}_1)}{(1-\delta)M^2}.$$

In particular, for any $\epsilon > 0$, choosing $M = \sqrt{\frac{2\,\mathrm{Var}(\widetilde{Z}_1)}{(1-\delta)\epsilon}}$ gives for large $n$,

$$\mathbb{P}\left(\max_{|k-n|\leq n\delta} \sqrt{k}\,|\lambda_k^\star - \lambda^\star| > CM\right) \leq \epsilon,$$

which implies $\max_{|k-n|\leq n\delta} \sqrt{k}|\lambda_k^\star - \lambda^\star| = O_p(1)$, proving $\sqrt{n}\max_{|k-n|\leq n\delta}|U_k| \to 0$ in probability.

**Case 3: $\Phi(q, m_o) = 1$.** In this case, $\lambda^\star = \bar{\lambda}$ and $\Psi(\lambda^\star, q) = 0$. Using Lemma A.6, we also have,

$$\lambda^\star - \lambda_k^\star = \frac{[\Psi(\lambda^\star, \hat{q}_k) - \Psi(\lambda^\star, q)]_+}{\Psi'(c_k, \hat{q}_k)}, \qquad [x]_+ := \max\{x, 0\}, \tag{46}$$

and $\Psi'(c_k, \hat{q}_k) \to \mathbb{E}[g'(\lambda^\star, X)] \in (0, \infty)$ a.s.

Now, recall Term 2,

$$\sqrt{n}|U_n| = \sqrt{n}\,|\lambda_n^\star - \lambda^\star|\,|A_n|.$$

By Lemma A.6 and Lemma A.7, we have $\sqrt{n}|\lambda_n^\star - \lambda^\star| = O_p(1)$, and $A_n \to 0$ a.s., hence $\sqrt{n}|U_n| \to 0$ in probability.

Term 1: We already know that $A_k \to 0$ a.s., hence $\max_{|k-n|\leq n\delta}|A_k| \to 0$ a.s. It remains to show $\max_{|k-n|\leq n\delta} \sqrt{k}|\lambda_k^\star - \lambda^\star| = O_p(1)$. From (46) for all large $k$, using proofs similar to case 2, we have

$$\sqrt{k}|\lambda_k^\star - \lambda^\star| \leq C\sqrt{k}\big|\Psi(\lambda^\star, \hat{q}_k) - \Psi(\lambda^\star, q)\big| = C\left|\frac{1}{\sqrt{k}}\sum_{i=1}^k g(\lambda^\star, X_i) - \mathbb{E}[g(\lambda^\star, X)]\right|.$$

Similar to the proof of case 2, using Kolmogorov inequality, we get that Term 1, i.e., $\sqrt{n}\max_{|k-n|\leq n\delta}|U_k| \to 0$ in probability.

**Step 5: Handling term (A).** By (39) and the union bound,

$$\mathbb{P}\left(\sqrt{n}\max_{|k-n|\leq n\delta}|L_k - L_n| > \epsilon/2\right) \leq \mathbb{P}\left(\sqrt{n}\max_{|k-n|\leq n\delta}|U_k| > \epsilon/6\right) + \mathbb{P}\left(\sqrt{n}|U_n| > \epsilon/6\right)$$
$$+ \mathbb{P}\left(\sqrt{n}\max_{|k-n|\leq n\delta}|V_{k,n}| > \epsilon/6\right).$$

Recall that from step 4, we have $\sqrt{n}|U_n| \to 0$ in probability and $\sqrt{n}\max_{|k-n|\leq n\delta}|U_k| \to 0$ in probability. Therefore, for each fixed $\delta \in (0, 1)$ and each $\epsilon > 0$,

$$\lim_{n\to\infty} \mathbb{P}\big(\sqrt{n}|U_n| > \epsilon/6\big) = 0, \qquad \lim_{n\to\infty} \mathbb{P}\left(\sqrt{n}\max_{|k-n|\leq n\delta}|U_k| > \epsilon/6\right) = 0.$$

For the third probability, (43) yields, for every $\delta \in (0, 1)$ and every $n \geq 1$,

$$\mathbb{P}\left(\sqrt{n}\max_{|k-n|\leq n\delta}|V_{k,n}| > \epsilon/6\right) \leq C_V\delta + C_V'\delta^2.$$

Now fix $\epsilon > 0$ and $\eta > 0$, and choose $\delta \in (0, 1)$ such that

$$C_V\delta + C_V'\delta^2 < \eta/4.$$

With this choice of $\delta$, by the limits above there exists $n_0 = n_0(\delta, \epsilon, \eta)$ such that for all $n \geq n_0$,

$$\mathbb{P}\left(\sqrt{n} \max_{|k-n|\leq n\delta} |U_k| > \epsilon/6\right) < \eta/8, \qquad \mathbb{P}\left(\sqrt{n}|U_n| > \epsilon/6\right) < \eta/8.$$

Combining these bounds, we get that for all $n \geq n_0$,

$$\mathbb{P}\left(\sqrt{n} \max_{|k-n|\leq n\delta} |L_k - L_n| > \epsilon/2\right) < \eta/8 + \eta/8 + \eta/4 = \eta/2,$$

which proves (A).

**Step 6: Handling term (B).** Fix $\epsilon > 0$ and $\eta > 0$. Let $\delta \in (0, 1)$ be arbitrary (to be chosen later). Recall

$$M_{n,\delta} := \max_{|k-n|\leq n\delta} \sqrt{n}\,|L_k - L|.$$

For $|k - n| \leq n\delta$ we have $k \in [(1 - \delta)n, (1 + \delta)n] \subset [1, 2n]$ for all large $n$. Moreover,

$$\sqrt{n}|L_k - L| \leq \sqrt{n}|U_k| + \sqrt{n}\left|\frac{1}{k}\sum_{i=1}^{k} \ell(\lambda^\star, X_i) - \mathbb{E}[\ell(\lambda^\star, X)]\right| = \sqrt{n}|U_k| + \sqrt{n}\left|\frac{S_k}{k}\right|,$$

where $S_k := \sum_{i=1}^{k} Z_i$ and $Z_i := \ell(\lambda^\star, X_i) - \mathbb{E}[\ell(\lambda^\star, X)]$ with $\mathbb{E}[Z_i] = 0$ and $\mathrm{Var}(Z_1) =: \sigma_Z^2 < \infty$. Hence,

$$M_{n,\delta} \leq \underbrace{\max_{|k-n|\leq n\delta} \sqrt{n}|U_k|}_{=:M_{n,\delta}^{(U)}} + \underbrace{\max_{|k-n|\leq n\delta} \sqrt{n}\left|\frac{S_k}{k}\right|}_{=:M_{n,\delta}^{(S)}}.$$

Therefore,

$$\mathbb{P}\left(M_{n,\delta} > \frac{\epsilon}{2\delta}\right) \leq \mathbb{P}\left(M_{n,\delta}^{(U)} > \frac{\epsilon}{4\delta}\right) + \mathbb{P}\left(M_{n,\delta}^{(S)} > \frac{\epsilon}{4\delta}\right). \tag{47}$$

For $k \in \mathbb{Z}^+$ such that $|k - n| \leq n\delta$ we have $k \geq (1 - \delta)n$, so for all large $n$,

$$\sqrt{n}\left|\frac{S_k}{k}\right| = \frac{n}{k}\left|\frac{S_k}{\sqrt{n}}\right| \leq \frac{1}{1-\delta}\left|\frac{S_k}{\sqrt{n}}\right| \leq \frac{1}{1-\delta} \max_{t\leq 2n}\left|\frac{S_t}{\sqrt{n}}\right|.$$

Hence, for any $a > 0$ and all large $n$,

$$\mathbb{P}\left(M_{n,\delta}^{(S)} > a\right) \leq \mathbb{P}\left(\max_{t\leq 2n} |S_t| > a(1 - \delta)\sqrt{n}\right).$$

By Kolmogorov's maximal inequality, for any $x > 0$ and any $n \geq 1$,

$$\mathbb{P}\left(\max_{t\leq 2n} |S_t| > x\right) \leq \frac{\mathrm{Var}(S_{2n})}{x^2} = \frac{2n\sigma_Z^2}{x^2}.$$

Applying this with $x = a(1 - \delta)\sqrt{n}$ yields, for all large $n$,

$$\mathbb{P}\left(M_{n,\delta}^{(S)} > a\right) \leq \frac{2\sigma_Z^2}{a^2(1 - \delta)^2}.$$

Now take $a = \epsilon/(4\delta)$. Then for all large $n$,

$$\mathbb{P}\left(M_{n,\delta}^{(S)} > \frac{\epsilon}{4\delta}\right) \leq \frac{2\sigma_Z^2}{(1 - \delta)^2} \cdot \frac{16\delta^2}{\epsilon^2} = \frac{32\sigma_Z^2}{\epsilon^2} \cdot \frac{\delta^2}{(1 - \delta)^2}. \tag{48}$$

Choose $\delta_0 \in (0, 1/2)$ such that for all $\delta \in (0, \delta_0)$,

$$\frac{32\sigma_Z^2}{\epsilon^2} \cdot \frac{\delta^2}{(1-\delta)^2} \leq \frac{\eta}{4}.$$

Then, for any such $\delta$, (48) gives for all large $n$,

$$\mathbb{P}\Big(M_{n,\delta}^{(S)} > \frac{\epsilon}{4\delta}\Big) \leq \frac{\eta}{4}. \tag{49}$$

In Case 1, $M_{n,\delta}^{(U)} = 0$ for all large $n$ a.s., so

$$\mathbb{P}\Big(M_{n,\delta}^{(U)} > \frac{\epsilon}{4\delta}\Big) = 0 \quad \text{for all large } n.$$

In Cases 2 and 3, we have $M_{n,\delta}^{(U)} \to 0$ in probability (Step 4). Hence, for the fixed $\delta \in (0, \delta_0)$ chosen above, there exists $N(\delta, \epsilon, \eta)$ such that for all $n \geq N(\delta, \epsilon, \eta)$,

$$\mathbb{P}\Big(M_{n,\delta}^{(U)} > \frac{\epsilon}{4\delta}\Big) \leq \frac{\eta}{4}. \tag{50}$$

*Combine.* Combining (47), (49), and (50), for any $\delta \in (0, \delta_0)$ and all $n \geq N(\delta, \epsilon, \eta)$,

$$\mathbb{P}\Big(M_{n,\delta} > \frac{\epsilon}{2\delta}\Big) \leq \frac{\eta}{4} + \frac{\eta}{4} = \frac{\eta}{2}.$$

This proves (B).

**Step 7: Conclusion.** Fix $\epsilon > 0$ and $\eta > 0$.

First, by (A) (proved in Step 4), choose $\delta_A \in (0, 1)$ and $n_A$ such that for all $n \geq n_A$,

$$\mathbb{P}\Big(\sqrt{n} \max_{|k-n| \leq n\delta_A} |L_k - L_n| > \epsilon/2\Big) < \eta/2.$$

Second, by (B) (proved in Step 5), choose $\delta_0 \in (0, 1/2)$ sufficiently small so that for some $n_B$ and all $n \geq n_B$,

$$\mathbb{P}\Big(M_{n,\delta_0} > \frac{\epsilon}{2\delta_0}\Big) < \eta/2.$$

Now set

$$\delta := \min\{\delta_A, \delta_0\}.$$

Since shrinking $\delta$ can only decrease the maxima over the set $\{|k - n| \leq n\delta\}$, the two bounds above remain valid (possibly with the same or smaller probabilities) when $\delta_A$ and $\delta_0$ are replaced by $\delta$. Hence, letting

$$n_1 := \max\{n_A, n_B\},$$

we have for all $n \geq n_1$,

$$\mathbb{P}\Big(\sqrt{n} \max_{|k-n| \leq n\delta} |L_k - L_n| > \epsilon/2\Big) < \eta/2, \qquad \mathbb{P}\Big(M_{n,\delta} > \frac{\epsilon}{2\delta}\Big) < \eta/2.$$

Therefore, using (37) and the union bound, for all $n \geq n_1$,

$$\mathbb{P}\Big(\max_{|k-n| \leq n\delta} |Y_k - Y_n| > \epsilon\Big) \leq \mathbb{P}\Big(\sqrt{n} \max_{|k-n| \leq n\delta} |L_k - L_n| > \epsilon/2\Big) + \mathbb{P}\Big(M_{n,\delta} > \frac{\epsilon}{2\delta}\Big) < \eta.$$

This completes the proof.

$\square$

## B. Discussion on Assumption 4.1

We now show that Assumption 4.1 is mild, by demonstrating that it holds across several commonly studied distribution families. In the discussion below, we focus on the case $m_o > m(q)$; the case $m_o < m(q)$ follows by the symmetric argument under the transformation $X \mapsto 1 - X$.

**Distributions with an atom at 1 (e.g., Bernoulli).** If $q(\{1\}) > 0$, then $\mathbb{E}_q\left[\frac{1-m_o}{1-X}\right] = \infty$ for every $m_o \in (0, 1)$. Hence, Assumption 4.1 is vacuously satisfied for such distributions, and in particular, for the entire Bernoulli family.

**Uniform distribution.** We note that $\mathrm{Uniform}[-1, 1]$ is not supported on $[0, 1]$; however, all the results in the paper can be adapted to the setting of distributions supported in $[a, b]$ for any $a < b$, via a rescaling argument. For simplicity of presentation, we consider $q = \mathrm{Uniform}[0, 1]$ here. For this,

$$\mathbb{E}_q\left[\frac{1-m_o}{1-X}\right] = (1 - m_o) \int_0^1 \frac{\mathrm{d}x}{1-x} = \infty.$$

Hence, Assumption 4.1 is vacuously satisfied for $\mathrm{Uniform}[0, 1]$.

The same argument applies for the **Truncated Normal** distribution on $[0, 1]$.

**Beta family.** For $q = \mathrm{Beta}(a, b)$ with $a, b > 0$, we have $q(\{1\}) = 0$ and

$$\mathbb{E}_q\left[\frac{1-m_o}{1-X}\right] = (1 - m_o)\frac{a+b-1}{b-1}, \qquad b > 1,$$

and $\infty$ for $b \leq 1$, so Assumption 4.1 holds vacuously for $b \leq 1$. For $b > 2$, $\mathbb{E}_q[1/(1-X)^2] < \infty$, so Assumption 4.1 holds trivially as well. It remains to consider $b \in (1, 2]$, where $\mathbb{E}_q\left[\frac{1-m_o}{1-X}\right] = 1$ if and only if

$$m_o = m_o^\star(a, b) := \frac{a}{a+b-1},$$

and at this $m_o$ one has $\mathbb{E}_q[1/(1-X)^2] = \infty$. Hence Assumption 4.1 is **violated for exactly one value of** $m_o$, namely $m_o^\star(a, b)$.

In summary, for a given $\mathrm{Beta}(a, b)$ distribution, Assumption 4.1 always holds except for a unique critical value of $m_o$ in the case $m_o > m(q)$. The case $m_o < m(q)$ follows symmetrically by applying the same reasoning to $1 - X$. Thus, this assumption holds for several commonly studied families of distributions with bounded support.

Hence, we believe Assumption 4.1 is a mild technical condition.

## C. Discussion on practical alternative of $\beta(n, \alpha)$

The choice $\beta(n, \alpha) = \log(1/\alpha)$ has both empirical and some theoretical support. The theoretical justification for the leading term comes from the null asymptotics of the $\mathrm{KL}_{\inf}$ statistic. This statistic can be viewed as a (non-parametric) log-likelihood ratio, and a version of Wilks' theorem can be proven for it. While this has been observed empirically earlier, this asymptotic was very recently established theoretically by Wang et al. (2026, Theorem 5.1). Specifically, the mentioned reference establishes that, under the null, $2n\,\mathrm{KL}_{\inf}(\hat{q}_n, m_o)$ converges weakly to a $\chi^2(1)$ distribution as $n \to \infty$. For small $\alpha$, the stopping time $\tau_\alpha$ is large, so this asymptotic regime is precisely the relevant one. By the $\chi^2(1)$ limit, controlling $P_{\mathrm{null}}(n\,\mathrm{KL}_{\inf}(\hat{q}_n, m_o) \geq \beta(n, \alpha)) \leq \alpha$ requires $2\beta(n, \alpha)$ to approximate the $(1 - \alpha)$ quantile of $\chi^2(1)$. Since $P(\chi^2(1) > x) \approx 2\Phi(-\sqrt{x})$ for large $x$, this quantile is approximately $2\log(1/\alpha)$ for small $\alpha$, yielding $\beta(n, \alpha) \approx \log(1/\alpha)$ as the natural leading term.

However, this $\chi^2$-based argument is inherently asymptotic in $n$ and therefore does not, by itself, guarantee type-I error control at any finite sample size or finite $\alpha$. The adjusted threshold $\beta(n, \alpha) = 1 + \log(\frac{2(1+n)}{\alpha})$ precisely compensates for this limitation. First, the additional terms correct for the asymptotic approximation and ensure that the above bound holds non-asymptotically, i.e., for all finite $\alpha$ and all sample sizes. Second, this correction also yields a *time-uniform* (in $n$) guarantee, meaning that the type-I error control holds simultaneously over all (possibly random) stopping times.

Empirically, we observe that the $\chi^2(1)$ approximation becomes accurate already at practically relevant levels such as $\alpha = 0.05$ and $\alpha = 0.01$, which further supports $\log(1/\alpha)$ as the dominant term governing the choice of $\beta(n, \alpha)$.

