# OpenReview forum: "Beyond First-order Asymptotics in Sequential Mean Testing"
_ICML.cc/2026/Conference — ICML 2026 spotlight_

### Official Review · Reviewer_Ejng · 2026-03-09

**Soundness:** 4
**Presentation:** 4
**Significance:** 4
**Originality:** 4
**Overall Recommendation:** 5
**Confidence:** 4

**Summary:**

The authors study the detection time in the mean testing problem (simple null vs simple alternative) in the setting where the random variables are bounded. Existing results in the literature mainly establish first-order optimality results expressed in terms of information-theoretic quantities. However, such results have important limitations: although the expected detection time may be small, it can still take very large values with non-negligible probability. In this paper, the authors focus on stronger second-order guarantees and develop a central limit theorem (CLT) for the detection time. The proof relies on a CLT for the relevant information-theoretic quantity, which the authors also establish.

**Compliance With Llm Reviewing Policy:**

Affirmed.

**Final Justification:**

Dear all, I appreciated the constructive discussions during the review of this paper. This is a very interesting and strong paper, and I recommend acceptance.

**Key Questions For Authors:**

1. Do we understand why the choice $\beta(n,\alpha)=\log(1/\alpha)$ works well? Are there theoretical results explaining this choice, or is it primarily supported by empirical evidence?

2. Can the results be extended to a regime different from the one where $\alpha \rightarrow 0$? If so, what would be the main differences in that setting?

**Limitations:**

yes

**Strengths And Weaknesses:**

**Strengths:**

1. The paper is theoretically very solid, and the proofs in the appendix are detailed and comprehensive. The proof sketch in the main paper is also very clear.

2. The paper is very well written. The only minor point I noted is in line 198: when the authors state that $\lambda$ maximizes $KL_{inf}(q,m_{0})$, I find the wording somewhat unclear. It would be more precise to explain that $\\lambda^*(q)$ denotes the value of $\lambda$ that achieves the supremum defining $KL_{inf}(q,m_0)$.

3. The paper presents a novel result that is significant, as it provides a more refined characterization of the detection times, going beyond first-order guarantees and offering deeper insight into their distributional behavior.

4. The experimental setup is clear and relevant.

**Weaknesses:**

The plots could be improved. In particular, the authors do not specify how the histograms were constructed: were they produced using a fixed number of bins or a fixed bin width? More importantly, it is unclear whether the “gaps” or “spikes” observed in the histograms in Figures 3 and 4 reflect a genuine statistical phenomenon or are simply plotting artifacts. The fonts used in the plots are also difficult to read, as they are too small.

---

> ### Author Rebuttal · Authors · 2026-03-31
>
> We thank Reviewer Ejng for their careful reading and constructive feedback. We address each question below.
>
> ---
>
> **Response to a minor point from Strengths.** We thank the reviewer for this suggestion. We agree that the wording was imprecise. In the revised version, we clarify that $\lambda^\star(q)$ denotes the value of $\lambda$ that achieves the supremum in the definition of $\mathrm{KL}_{\inf}(q, m_0)$.
>
> ---
>
>
> **Response to Weaknesses.** We thank the reviewer for these helpful comments. In the current version, the histograms are constructed using a fixed number of bins. The gaps and spikes visible in Figures 3 and 4 are plotting artefacts and do not reflect any genuine statistical phenomenon. We have specified the histogram construction explicitly in the captions and increase the font sizes throughout all figures.
>
> ---
>
> **Answer to Key Question 1.** The choice $\beta(n, \alpha) = \log(1/\alpha)$ has both empirical and some theoretical support. The theoretical justification for the leading term comes from the null asymptotics of the $\operatorname{KL_{inf}}$ statistic. This statistic can be viewed as a (non-parametric) log-likelihood ratio, and a version of Wilks' theorem can be proven for it. While this has been observed empirically earlier, this asymptotic was very recently established theoretically in Theorem 5.1 of [1].
>
> Specifically, the mentioned reference establishes that, under the null,  $2n \operatorname{KL_{inf}}(\hat{q}_n, m_0)$
>
> converges weakly to a $\chi^2(1)$ distribution as $n \to \infty$. For small $\alpha$, the stopping time $\tau_\alpha$ is large, so this asymptotic regime is precisely the relevant one. By the $\chi^2(1)$ limit, controlling $P_{\mathrm{null}}(n \operatorname{KL_{inf}}(\hat{q}_n, m_0) \geq \beta(n,\alpha)) \leq \alpha$ requires $2\beta(n,\alpha)$ to approximate the $(1-\alpha)$ quantile of $\chi^2(1)$. Since $P(\chi^2(1) > x) \approx 2\Phi(-\sqrt{x})$ for large $x$, this quantile is approximately $2\log(1/\alpha)$ for small $\alpha$, yielding $\beta(n,\alpha) \approx \log(1/\alpha)$ as the natural leading term.
>
> However, this $\chi^2$-based argument is inherently asymptotic in $n$ and therefore does not, by itself, guarantee type-I error control at any finite sample size or finite $\alpha$. The adjusted threshold $\beta(n, \alpha) = 1 + \log(\frac{2(1+n)}{\alpha})$, given below Eq. (3), precisely compensates for this limitation. First, the additional terms correct for the asymptotic approximation and ensure that the above bound holds non-asymptotically, i.e., for all finite $\alpha$ and all sample sizes. Second, this correction also yields a *time-uniform* (in $n$) guarantee, meaning that the type-I error control holds simultaneously over all (possibly random) stopping times.
>
> Empirically, we observe that the $\chi^2(1)$ approximation becomes accurate already at practically relevant levels such as $\alpha = 0.05$ and $\alpha = 0.01$, which further supports $\log(1/\alpha)$ as the dominant term governing the choice of $\beta(n,\alpha)$.
>
> ---
>
>
> **Answer to Key Question 2.** Recall that we focus on a composite-vs-composite hypothesis testing problem. In that setting, beyond the $\alpha \to 0$ regime studied in this paper, the other natural asymptotic regime is $\operatorname{KL_{inf}}(q, m_o) \to 0$, for a fixed $\alpha$. This can be achieved by considering a sequence of alternative data-generating distributions, whose mean approaches $m_0$. In this regime, the expected stopping time has been recently shown to grow as,  $ \Omega \left( \operatorname{KL_{inf}}^{-1} \log\log \operatorname{KL_{inf}}^{-1} \right)$, a rate reminiscent of the law of the iterated logarithm (see [2] for more details). The reference also shows some tests achieving this rate. However, even the first order asymptotics in this regime are not completely understood in the non-parametric setting. We therefore leave this as an interesting direction for future work.
>
> We thank the reviewer for raising this question. We have now included this discussion in our Conclusions and future work section.
>
> ---
>
> [1] Wang, H., Agrawal, S. and Ramdas, A. (2026). Almost sure null bankruptcy of testing-by-betting strategies. arXiv preprint arXiv:2602.08888.
>
> [2] Agrawal, S. and Ramdas, A. (2025). On stopping times of power-one sequential tests: Tight lower and upper bounds. arXiv preprint arXiv:2504.19952.

---

> > ### Author Rebuttal · Reviewer_Ejng · 2026-04-02
> >
> > I thank the authors for their response. I have also read the other reviews. I believe this paper is strong and well-executed, and I am happy to maintain my score of 5.

---

> > > ### Author Response · Authors · 2026-04-05
> > >
> > > We once again thank the reviewer for the careful reading and for acknowledging our clarifications. We appreciate the constructive feedback, which helped improve the quality and clarity of our work.

---

### Official Review · Reviewer_QmRu · 2026-03-10

**Soundness:** 3
**Presentation:** 3
**Significance:** 3
**Originality:** 3
**Overall Recommendation:** 5
**Confidence:** 2

**Summary:**

In this paper, the authors are interested in a mean testing problem, where the null hypothesis tests whether a distribution of interest (denoted by $p$) has an expectation of $m_0$, while the alternative hypothesis consists of distributions that have an expectation different from $m_0$. With this test in mind, they consider the stopping time of the test (denoted by $\tau_\alpha$) at confidence level $\alpha$. Then, their paper is based on a well-known link between the expected stopping time and an optimization problem that finds the distribution $p$ minimizing a Kullback–Leibler divergence between $p$ and a reference distribution (denoted by $q$), subject to a constraint on the expectation with respect to $m_0$; this quantity is called in the paper the KL projection function $\mathrm{KL}_{\mathrm{inf}}$. Then, the authors propose a central limit theorem on a (scaled) difference between a KL projection function of an empirical distribution and the true distribution. Then, they link the stopping time and the KL projection with a similar approach as in the literature but in probability, which allows them to obtain a central limit theorem on the stopping time.

**Compliance With Llm Reviewing Policy:**

Affirmed.

**Final Justification:**

The authors have correctly addressed my questions.

**Key Questions For Authors:**

- Why the condition $\mathbb{E}\left[1/(1-X)^2\right]$ is needed? Is it restrictive?
- Why the parameters for experiments 1 and 2 of the Bernoulli change? Do the convergences change for these two parameters? (if so, can we have empirical evidence?)

**Limitations:**

yes

**Strengths And Weaknesses:**

**Strengths.**

- While I am not an expert in this area, I find the paper easy to follow; I still have some minor remarks.
- While being theoretical work, the authors provide empirical evidence of the central limit theorems.

**Weaknesses.**

- The presentation of the paper can be improved (while still easy to follow and pleasant to read).
- While I like the fact that the authors provide some experiments for this theoretical work, I still have two points that I find not convincing
    - The parameters of the Bernoulli between the Experiments 1 and 2 change for no apparent reasons;
    - While one of the selling point of the paper is to the fact that the test holds for unknown distribution, the experiments are on well known distributions. I would have been happy to see results on a real-world dataset.

**Remarks.**

I have some remarks concerning the clarity of the paper and the notations.

- I find the notations $p(\tau_\alpha < \infty)$ and $q(\tau_\alpha < \infty)$ weird in the sense that $p(\cdot)$ measures a probability event on a single realization $X\sim p$ and $X\sim q$ while $\tau_\alpha$ is a random variable that depends on the sequence $X_1,\dots$. Consider writing something like $\mathbb{P}_{X_1 \sim p, \dots}$ in the notation.
- Similarly, the expectation $\mathbb{E}q[X]$ and $\mathbb{q}[\tau\alpha]$ are not of the same nature, i.e., the second one depends on the sequence while the first one depends on a single realization: the notation must distinguish the two for clarity; my comment also applies to the probability $\mathbb{P}_{q}$.
- What is the notation $O_p(\cdot)$?
- The arrow $\Rightarrow$ for the convergence in distribution is confusing because you use another arrow for the other convergences; moreover, these convergence notations are not introduced in the main paper; consider adding in the notations.
- The acronym SLLN is not introduced if I'm not mistaken.
- The year of a citation is missing (one of the papers of Deep et al.); moreover, the style of the references is not unified; consider cleaning it.
- The figures are unreadable: (i) the font is too small, (ii) it is difficult to read in black and white; consider changing the colors and increase the font size.
- The parentheses near the log functions are too close.

---

> ### Author Rebuttal · Authors · 2026-03-31
>
> We thank the reviewer for their careful reading and constructive feedback. We address each question below.
>
> ---
>
> **Answer to Weakness 1.** In the revised version, we have gone over the paper carefully and incorporated the suggestions raised. In particular, we have fixed the typos, notation, and improved the exposition throughout. We hope this makes the paper more concise and easier to read.
>
> ---
>
> **Answer to Weakness 3.** We thank the reviewer for this suggestion and are happy to report that we have conducted an additional experiment on a real-world dataset, the results of which have been added to the paper. The data originates from the DSSAT simulator, an open-source, physics-based crop-growth model widely used in agronomy research and maintained by the DSSAT Foundation (https://dssat.net). Specifically, we borrow the cleaned crop-yield dataset from the supporting material of [1], and we acknowledge the authors of [1] in our paper for cleaning the data and making it publicly available. Each treatment option corresponds to a distinct combination of planting date and fixed soil conditions. The dataset consists of $1{,}000{,}000$ observations, which we scale to $[0,1]$ to make it compatible with our bounded distribution setting. We set the null hypothesis value $m_0 = 0.5$ and run our sequential test with significance level $\alpha = 10^{-4}$. To simulate the stopping time distribution, we generate $3{,}000$ independent bootstrap paths by resampling with replacement from the data pool.
>
> Since the underlying distribution $q$ is unknown, $\frac{1}{\operatorname{KL_{inf}}(q,m_o)}$ and $\sigma^2_{\mathrm{bd}}$ are not available in closed form; we instead substitute plug-in estimates from the data, $\frac{1}{\operatorname{KL_{inf}}(\hat{q},m_o)}$ and $\hat{\sigma}^2_{\mathrm{bd}}$. We find that the empirical distribution of $\frac{\tau_\alpha}{\log(1/\alpha)}$ closely matches the $\mathcal{N}\left(\frac{1}{\operatorname{KL_{inf}}(\hat{q},m_o)}, \hat{\sigma}^2_{\mathrm{bd}}\right)$ overlay, providing further empirical support for our asymptotic CLT result in a real-world setting.
>
> **References**
>
> [1] Jourdan, M., Degenne, R., Baudry, D., De Heide, R. and Kaufmann, E. Top Two Algorithms Revisited. *Advances in Neural Information Processing Systems (NeurIPS)*, 2023.
>
> ---
>
> **Answer to R1 and R2.** In the revised version, we now use the notation $\mathbb{P}_q(\cdot)$ (and $\mathbb{P}_p(\cdot)$) to denote the probability of the input event under the measure on an infinite length sequence $X_1, X_2, X_3, \ldots$, generated i.i.d. from $q$ (and $p$). Similarly, we now use $\mathbb{E}_q[\cdot]$ (and $\mathbb{E}_p[\cdot]$) to denote the expectation under this joint measure. We define this notation early on in the revised manuscript.
>
> ---
>
>
> **Answer to R3.** $O_p(\cdot)$ is a standard notation that denotes *stochastic boundedness*: a sequence of random variables $(Z_n)$ satisfies $Z_n = O_p(a_n)$ if, for every $\varepsilon > 0$, there exist $M_\varepsilon, N_\varepsilon < \infty$ such that $\mathbb{P}(|Z_n/a_n| > M_\varepsilon) < \varepsilon$ for all $n \geq N_\varepsilon$. We have explicitly defined this notation in the revised paper.
>
> ---
>
> **Answer to R4.** We have added a dedicated notation paragraph at the start of the paper that defines all convergence modes used: $\xrightarrow{\mathrm{a.s.}}$ for almost sure convergence, $\xrightarrow{p}$ for convergence in probability, and $\xrightarrow{d}$ for convergence in distribution. All these symbols are used consistently throughout.
>
> ---
>
> **Answer to R5.** We have now defined it at its first occurrence in the revised paper.
>
> ---
>
> **Answer to R6.** Thank you for catching this. We have added the missing publication year to the Deep et al. reference and have gone through the full reference list to unify the citation style throughout.
>
> ---
>
>
> **Answer to R7.** Thank you for this observation. We have re-run all experiments and reproduced the figures with larger font sizes and a colour scheme that is legible both in colour and in greyscale. The updated figures appear in the revised paper.
>
> ---
>
>
> **Answer to R8.** We have fixed the tight parentheses so the spacing is now visually correct.
>
> ---
>
> **Answer to Key Q1.** Thank you for raising this question. We refer the reviewer to our detailed response under "Weaknesses and Q1" in the response to Reviewer GJ8H.
>
> ---
> **Answer to Weakness 2 and Key Q2.** In the revised version, both experiments use a Bernoulli$(0.6)$ distribution for consistency. The updated figures are included in the revision.

---

> > ### Author Rebuttal · Reviewer_QmRu · 2026-04-01
> >
> > The authors have correctly addressed my questions. Hence, I have increased my overall score by 1 point.

---

> > > ### Author Response · Authors · 2026-04-05
> > >
> > > We once again thank the reviewer for the careful reading, for increasing the score, and for the valuable feedback.

---

### Official Review · Reviewer_GJ8H · 2026-03-12

**Soundness:** 3
**Presentation:** 2
**Significance:** 2
**Originality:** 2
**Overall Recommendation:** 4
**Confidence:** 2

**Summary:**

The authors consider the sequential mean testing problem, which is a hypothesis test with type I error $\alpha$ and power $1$. In the limit $\alpha \to 0$, it is shown that the fluctuation of the stopping time of such a test converges to a Gaussian, where the mean and the variance of the limiting Gaussian are also explicitly given. Results from numerical experiments are provided.

**Compliance With Llm Reviewing Policy:**

Affirmed.

**Final Justification:**

The authors answered my question adequately. I changed the score accordingly.

**Key Questions For Authors:**

- My main concern is the assumption $E_q[1/(1-X)^2] < \infty$. If my understanding is correct, it excludes any distribution with an atom at $1$ (e.g., Bernoulli distribution) or with density that does not decay fast enough near $1$ (e.g., uniform distribution on [-1, 1]). Is the assumption just a technical one?
- It seems that the distributions used in the numerical experiments, $q \sim Beta(3, 2)$, $q \sim Bernoulli(0.6)$, and $q \sim Bernoulli(0.7)$ do not satisfy the assumption $E_q[1/(1-X)^2] < \infty$. Thus, I wonder whether the numerical experiments are reasonable.

**Limitations:**

Yes

**Strengths And Weaknesses:**

Strengths
- Mathematical analysis is sound.
- Presentation is good in general.
- The result about the Gaussian convergence of the stopping time is new and can be potentially applied to other related problems.

Weaknesses
- The assumption is too strong; the random variable $X$ is not only bounded but also satisfy that $E_q[1/(1-X)^2] < \infty$, which is not satisfied by many distributions. (See also the question below.)

---

> ### Author Rebuttal · Authors · 2026-03-31
>
> We thank the reviewer for carefully reading the manuscript and providing a constructive feedback. We agree that CLT for the stopping time and $\operatorname{KL_{ inf}}$ are of independent interest, and could be applied for problems beyond the specific hypothesis testing framework considered in this work.
>
> ---
>
> ## Weaknesses and Q1
>
> We thank the reviewer for raising this. We clarify that the condition $\mathbb{E}_q\left[1/(1-X)^2\right] < \infty$ is **not** a general requirement for Theorem 4.1. It only arises in one of the cases in the proof. We have now fixed this in the theorem statement, as discussed below.
>
> **Assumption 1 (A1).** For $q\in\mathcal{B}$ and $m_o\in(0,1)$ with $m_o\neq m(q)$, if $\mathbb{E}_q \left[\frac{1-m_o}{1-X}\right] =1$, then $\mathbb{E}_q\left[1/(1-X)^2\right] < \infty$.
>
>  **Theorem 4.1.**  Fix $q\in\mathcal{B}$ and $m_o\in(0,1)$ with $m_o\neq m(q)$. Let $\hat q_n=\tfrac1n\sum_{i=1}^n \delta_{X_i}$ be the empirical distribution of i.i.d. $X_i\sim q$. Then, under Assumption 1,
> $$
> \sqrt{n}\Big( \operatorname{KL_{inf}}(\hat q_n,m_o) - \mathrm{KL}_{\inf}(q,m_o)\Big) \xrightarrow{d} \mathcal{N}\big(0,\sigma^2(q,m_o)\big),
> $$
> where $\sigma^2(q,m_o) = \operatorname{Var}_q \big(\ell(\lambda^\star,X)\big) < \infty$.
>
> Notice that A1 is much weaker than the previously stated assumption. To see that A1 suffices, recall from Section 4.1 that the proof for Theorem 4.1 distinguishes three cases based on the value of the following function:
>
> $$
> \Phi(q, m_o) := \mathbb{E}_{q} \left[\frac{1-m_o}{1-X}\right], \quad \text{with} \quad \Phi(q,m_o) = +\infty \text{ whenever } q(\{1\}) > 0.
> $$
>
> Let $\lambda^\star$ be the optimizer for $\operatorname{KL_{inf}}(q ,m_0)$. Then the three cases correspond to the following:
>
> - **Case 1.** $\Phi(q,m_o) < 1$: $\quad \lambda^\star = \bar\lambda := \tfrac{1}{1-m_o}$.
> - **Case 2.** $\Phi(q,m_o) > 1$: $\quad \lambda^\star < \bar\lambda$ solves $\Psi(\lambda^\star, q) = 0$.
> - **Case 3.** $\Phi(q,m_o) = 1$: $\quad \lambda^\star = \bar\lambda$ and $\Psi(\lambda^\star, q) = 0$.
>
> In our proof, the condition $\mathbb{E}_q\left[1/(1-X)^2\right] < \infty$ is needed only in **Case 3** to control the variance term in the CLT analysis; Cases 1 and 2 do not require this. We thank the reviewer for questioning this assumption. We have fixed the assumption and the statement of Theorem 4.1 as above, and have also modified all subsequent theorem and lemma statements accordingly. Note that this change does not affect any of our proofs.
>
> ---
>
> A1 is, in fact, mild. It holds across several commonly studied distribution families.
>
> 1. If $q(\{1\}) > 0$, then $\mathbb{E}_q\left[\frac{1-m_o}{1-X}\right] =\infty$ for every $m_o\in(0,1)$. Hence, (A1) is vacuously satisfied for such distributions, and in particular, for the Bernoulli family.
>
> 2. Note that $\mathrm{Uniform}[-1,1]$ is not supported on $[0,1]$, however all the results in the paper can be adapted to distributions supported in $[a,b]$ via a rescaling argument. For simplicity, we consider $q = \mathrm{Uniform}[0,1]$ here. For this,
> $$
> \mathbb{E}_q\left[\frac{1-m_o}{1-X}\right] = (1-m_o)\int_0^1 \frac{\mathrm{d}x}{1-x} = \infty.
> $$
> Hence, A1 is vacuously satisfied.
>
> 3. The same argument as above applies for the **Truncated Normal** distribution on $[0,1]$.
>
> 4. For $q=\mathrm{Beta}(a,b)$ with $a,b>0$, we have $q(\{1\})=0$ and
> $$
> \mathbb{E}_q\left[\frac{1-m_o}{1-X}\right] = (1-m_o)\,\frac{a+b-1}{b-1}, \qquad b>1,
> $$
> and $\infty$ for $b\le 1$, so A1 holds vacuously for $b\le 1$. For $b>2$, $\mathbb{E}_q[1/(1-X)^2]<\infty$, so A1 holds trivially as well. It remains to consider $b\in(1,2]$, where $\mathbb{E}_q\left[\frac{1-m_o}{1-X}\right]=1$ if and only if
> $$
> m_o = \frac{a}{a+b-1},
> $$
> and at this $m_o$ one has $\mathbb{E}_q[1/(1-X)^2]=\infty$. Hence A1 is **violated for exactly one value of $m_o$.
>
> In summary, for a given $\mathrm{Beta}(a,b)$, A1 always holds except for a unique value of $m_o$. Thus, A1 holds for several commonly studied families of distributions with bounded support. Hence, we believe A1 is a mild technical condition. We have now included a brief discussion along with these examples around A1 in the revised manuscript.
>
> ---
>
> ## Q2
>
>
>  We now verify that for all three distributions used for numerical experiments, Assumption 1 holds.
>
> - **Bernoulli$(p)$, $p\in\{0.6,0.7\}$:** Since $q(\{1\})=p>0$, we have $\mathbb{E}_q\left[\frac{1-m_o}{1-X}\right]=\infty$ for any $m_o \in (0,1)$, hence A1 vacuously holds.
>
> - **Beta$(3,2)$ with $m_o=0.7$:** Using the closed form from Answer 1,
> $$
> \mathbb{E}_q\left[\frac{1-0.7}{1-X}\right] = 1.2 > 1,
> $$
> hence A1 vacuously holds.
>
> All three distributions therefore satisfy the hypotheses of Theorem 4.1 as restated above, confirming that the numerical experiments are well-founded and consistent with the theory.

---

> > ### Author Rebuttal · Reviewer_GJ8H · 2026-04-01
> >
> > The authors answered my question adequately. I will adjust my score accordingly.

---

> > > ### Author Response · Authors · 2026-04-06
> > >
> > > We thank the reviewer for the careful reading, particularly regarding the paper's assumptions, for acknowledging our clarifications, and for increasing the score.

---

### Official Review · Reviewer_Y8Tg · 2026-03-18

**Soundness:** 4
**Presentation:** 3
**Significance:** 4
**Originality:** 3
**Overall Recommendation:** 5
**Confidence:** 3

**Summary:**

This paper studies the sequential mean testing model, where the goal is to define a stopping rule that determines if the mean of the underlying distribution from which the sequence arrives is m_0 or not, up to an error parameter alpha. Their main and novel contribution is to define a novel sequential test and then prove a central limit theorem for it.

**Compliance With Llm Reviewing Policy:**

Affirmed.

**Key Questions For Authors:**

Could you explain Lemma 4.4 and its proof in words and give a bit of intuition? I follow the results till then, but am not completely sure about the technical details of the Lemma.

**Limitations:**

Yes

**Strengths And Weaknesses:**

Strengths -
1. This work advances the current state of knowledge significantly. From the paper, it appears that it is the first study going beyond first-order results.
2. The experiments are naturally motivated and support the theoretical results.

Weaknesses -

1. The work is comprehensive but the writing is not very reader friendly. I would appreciate it if the authors give a more readable overview in the introduction, maybe defining a concrete toy problem.
2. The code for the experiments is not supplied for verification and better understanding.

---

> ### Author Rebuttal · Authors · 2026-03-31
>
> We thank the reviewer for their careful reading and constructive feedback. We address each question below.
>
> ---
>
> **Weakness 1.** We thank the reviewer for this suggestion. We have streamlined the introduction for better exposition. Further, we have added the motivation for studying the second-order asymptotic analysis of the sequential optimal tests in the context of sequential monitoring of safety metrics (e.g., extreme losses or excess risk in financial systems), more explicitly.
>
> ---
>
> **Weakness 2.** We thank the reviewer for this comment. We apologize for not submitting the code alongside the original submission. We will attach the code for all numerical experiments as a supplementary file in the revised submission.
>
> ---
>
>
> **Answer to Key Question.** Lemma 4.4 serves as an application of the stopping-time CLT (Theorem 4.3). It constructs an asymptotically valid confidence interval for  $1/\operatorname{KL_{inf}}(q,m_0)$,  or equivalently, for the rescaled expected stopping time $\mathbb{E}[\tau_\alpha]/\log(1/\alpha)$ for small $\alpha$, since these two quantities coincide in the limit $\alpha \to 0$.
>
> This is achieved by replacing the unknown asymptotic variance $\sigma^2_{\mathrm{bd}}(q,m_0)$ with a consistent plug-in estimator $\hat{v}_\alpha$.
>
> Concretely, the estimator
> $ \hat{v_\alpha} =  \hat{\sigma}^2_{\tau_\alpha} / L_{ \tau_{\alpha}}^3$
> satisfies
> $$ \hat{v_\alpha} \operatorname{\overset{p}{\to}} \sigma^2_{\operatorname{bd}}(q,m_0) $$
>
>  Plugging $\hat{v_\alpha}$ into the Gaussian limit of Theorem 4.3 via Slutsky's theorem yields the confidence interval $\mathcal{I}_\alpha(\gamma) $ defined around lines 312-314 (right),
>
> which is an asymptotically valid $(1-\gamma)$ confidence interval for $\mathbb{E}[\tau_\alpha]/\log(1/\alpha)$. This corresponds to the last display in Lemma 4.4. Informally, at the stopping time $\tau_\alpha$, $\mathcal{I}_\alpha(\gamma) $
>
> is an interval (constructed using the CLT) that contains $\frac{\mathbb{E}[\tau_\alpha]}{\log(1/\alpha)}$ for small $\alpha$ with probability $1-\gamma$, giving a practically usable interval even though $q$ is unknown. We have included this discussion after Lemma 4.4.
>
> **Note:** the appearance of $\delta$ inside the square root in the definition of $\mathcal{I}_\alpha(\gamma)$ in Lemma 4.4 is a typo. It should read as $\alpha$, consistent with the notation used throughout.

---

> > ### Author Rebuttal · Reviewer_Y8Tg · 2026-04-04
> >
> > Thank you for the rebuttal.

---

> > > ### Author Response · Authors · 2026-04-06
> > >
> > > We again thank the reviewer for the careful reading of the paper and constructive feedback.

---

### Official Review · Reviewer_yK9z · 2026-03-20

**Soundness:** 3
**Presentation:** 3
**Significance:** 3
**Originality:** 3
**Overall Recommendation:** 5
**Confidence:** 3

**Summary:**

The paper studies sequential testing of whether the mean of a distribution supported on $[0,1]$ takes a given value, based on an i.i.d. sample from that distribution. More specifically, it focuses on sequential tests based on $KL_{\inf}$, which have been shown to be optimal in several respects. In sequential testing, the quantity of interest is the stopping time $\tau_\alpha$, which, for a given level $\alpha$, corresponds to the time at which the null hypothesis can be rejected at level $\alpha$. For $KL_{\inf}$-based sequential testing, the asymptotic behaviour of the expected stopping time as $\alpha \to 0$ is known. However, this is often not sufficient, since $\tau_\alpha$ may still be very large with positive probability even when its expectation is small. In this paper, the authors establish a CLT for the stopping time, thereby improving our understanding of the fluctuations of $\tau_\alpha$ around its expected value.

**Compliance With Llm Reviewing Policy:**

Affirmed.

**Ethical Review Concerns:**

No concerns

**Final Justification:**

As reflected in my score I recommended this paper for acceptance. The rebuttal addressed my main concerns and I have updated my evaluation.

**Key Questions For Authors:**

# Main Comments

* In Theorems 4.1 and 4.3, it would be helpful to discuss the assumption $\mathbb{E}(1/(1-X)^2) < \infty$. Is this condition satisfied by a large class of distributions? For example, it seems that the uniform distribution does not satisfy it.
* L.246, right: “The maximizer $\lambda^*$ satisfies:”. A brief explanation in the appendix of why this are the relevant cases, along with a reference to that discussion in the main text, would make the paper more self-contained. At present, this appears to be stated without justification.
* L.319, left: “The results in Lemmas B.1 and B.2 states the maximizer ... almost sure converges to $\lambda^*$.” This sentence seems poorly constructed and should be rewritten.
* It is unclear to me why the alternative expression for $\beta(n,\alpha)$ yields an $\alpha$-correct stopping rule.
* It may also be useful to provide a confidence interval for $\tau_\alpha$.

# Small Comments

* It seems that the paragraph starting at L.073 on the left says almost the same thing as the beginning of the paragraph at L.088 on the left. If the statements are indeed the same, the repetition may be unnecessary. If there is an intended distinction, it should be made clearer.
* The legends in all figures are too small. More generally, the figures themselves are too small. It might be preferable to include only the plots for (n=1000) and (n=2000) in the main text.

# Tiny Comments

* In the introduction, the citation style with parentheses inside parentheses does not seem correct. Perhaps $\backslash citealp$ could be used where appropriate, as appears to be done elsewhere in the paper.
* L.393 and L.438, right: the journal name is missing capitalization.

**Limitations:**

Yes

**Strengths And Weaknesses:**

# Soundness
The submission seems technically sound: the claims appear to be well supported, and the relevant literature is properly discussed. The experiments are well designed and effectively illustrate the theoretical results. As mentioned below, I have some mild concerns about one assumption that is not discussed and about one result that should be supported by a citation, but these issues seem minor and could easily be addressed during the rebuttal.

# Presentation
The presentation is the main weakness of the paper, as there is considerable repetition. The proof of the main result is presented three times: first as a sketch in the introduction, then as a more detailed sketch in the main body, and finally in full in the appendix. Similarly, the technical difficulties involved in the proof are emphasized several times, which feels unnecessary. The paper would likely benefit from a rewrite aimed at reducing this repetition. For example, the proof sketch in the introduction could probably be omitted, which would free up space to better highlight the result mentioned in line 319. According to the authors, this is one of the paper’s main contributions, but it is not emphasized enough.

# Significance
The submission advances the understanding of sequential testing, which is widely used across a variety of fields.

# Originality
The submission provides new insights by improving the understanding of a method that has been proven optimal. The authors also develop a novel proof technique that could have broader impact by helping establish CLT results in complex settings.

---

> ### Author Rebuttal · Authors · 2026-03-31
>
> We thank the reviewer for their careful reading and constructive feedback. We address each question below.
>
> ---
>
> **Answer to Presentation.** We thank the reviewer for pointing this out. In the revised manuscript, we have appropriately reduced the discussion of the proof sketch and technical difficulties from the introduction and have added the following discussion on Lemma 4.4 (result mentioned in line 319).
>
> The objective of developing the CLT of $\tau_\alpha$ is to provide an asymptotically valid confidence interval of the stopping time, which can be reported along with the stopping time. Typically, confidence intervals in simulations are calculated using multiple independent simulation runs. Lemma 4.4 essentially aims to provide an asymptotically valid confidence interval of the scaled stopping time with a given coverage, using *a single run* of the simulation. It is worth noting that this Lemma relies on the CLT for scaled $\tau_\alpha$ developed in this work (Theorem 4.3).
>
> ---
>
>
> **Main Comment 1.** Thank you for raising this question. We refer the reviewer to our detailed response under "Weaknesses and Q1" in the response to Reviewer GJ8H.
>
> ---
>
> **Main Comment 2.** We thank the reviewer for pointing this out. The three cases arise because the dual representation of $\operatorname{KL_{inf}}(q, m_o)$ is a maximization of a concave function over a scalar parameter $\lambda$ constrained to a compact interval. Case 1 corresponds to the setting where the derivative of the objective is nonzero throughout the interval, and the maximum must therefore be attained at a boundary. In Case 2, the derivative of the objective is zero at a unique point in the interior. In Case 3, the derivative of the objective is zero at the boundary point of the interval. We have added a brief explanatory remark in the appendix at the relevant location and included a forward reference to it from the main text.
>
> ---
>
> **Main Comment 3.** We agree, and have rewritten the sentence as follows: "Lemmas B.1 and B.2 establish that the maximizer  $\lambda^{*}_{n} $
>
> of the optimization problem in the dual of $\operatorname{KL_{inf}}(\hat{q}_n, m_o)$ converges almost surely to $\lambda^\star$."
>
> ---
>
> **Main Comment 4.** The reviewer is correct: the alternative expression for $\beta(n,\alpha)$ does *not* come with a guarantee of $\alpha$-correctness, as stated in the paper. But it is a common practical threshold used in the literature. This expression is motivated empirically and enjoys some asymptotic theoretical support. We have discussed this in detail in our response to Reviewer Ejng, under "Answer to Key Question 1". We have now clarified this in the revised manuscript.
>
> ---
>
> **Main Comment 5.** We thank the reviewer for this helpful suggestion. We now provide two confidence intervals for $\tau_\alpha$ in the updated paper. First, using Lemma 4.4, we construct an asymptotically valid confidence interval derived from a *single* sample path, requiring no repeated experiments. Second, we also report an empirical confidence interval obtained directly from the observed histogram of the rescaled stopping times across the $3{,}000$ bootstrap paths. Together, these two intervals are reported in the updated numerical experiments, and we observe that they get closer as $\alpha$ shrinks, further corroborating the asymptotic theory.
>
> ---
>
> **Small Comment 1.** In the revised manuscript, we have merged them into a single, consolidated paragraph.
>
> ---
> **Small Comment 2.** We agree. In the revised manuscript we have enlarged all figures, increased the font size of all legends and axis labels, and retained only the plots for $n=1000$ and $n=2000$ in the main text. The remaining plots have been moved to the appendix.
>
> ---
> **Tiny Comments..** We thank the reviewer for catching these. We have replaced the relevant `\citep` commands with `\citealp` throughout the introduction to resolve the double-parenthesis issue, and have corrected the capitalisation of the journal names at L.393 and L.438.

---

> > ### Author Rebuttal · Reviewer_yK9z · 2026-04-01
> >
> > The authors have answered my questions. I'm sure they will make good use of the extra page to refine the presentation. I will adjust my score accordingly.

---

> > > ### Author Response · Authors · 2026-04-06
> > >
> > > We thank the reviewer for the careful reading, increasing the score, and the comments on presentation, which helped enhance the quality and clarity of the paper.

---

### Decision · Program_Chairs · 2026-04-30

**Decision:**

Accept (spotlight)

**Comment:**

All reviewers agree that this paper provides novel and useful theoretical results related to the sequential mean testing problem. In particular, it derives a CLT result for the stopping time in the small-error regime. The paper is well-written, mathematically strong and the results significant.
Please revise the paper as suggested in the responses to the reviewers, and provide the code for the numerical experiments.